# Evaluation of the hoof centre-of-pressure path in horses affected by chronic osteoarthritic pain

**Larissa Irina Buser**[1]*, **Nathan Torelli**[2], **Sabrina Andreis**[1], **Stefan Witte**[3⊙],
**Claudia Spadavecchia**[1⊙]

1 Vetsuisse Faculty, Department of Clinical Veterinary Medicine, Section of Anaesthesiology and Pain Therapy, University of Bern, Bern, Switzerland, 2 Department of Radiation Oncology, University Hospital Zürich and University of Zürich, Zürich, Switzerland, 3 Tierklinik Schönbühl AG, Schönbühl, Switzerland

⊙ These authors contributed equally to this work.
* larissa.buser@unibe.ch

## Abstract

**Data Availability Statement:** All relevant data are within the manuscript and supporting information files.

### Introduction

The Centre of Pressure (COP) is the single point summarising all forces transferred to the hoof during the stance phase of a stride. COP path (COPp) is the trajectory that COP follows from footstrike to lift-off. Aim of the present study was to characterize the COP and COPp in horses affected by osteoarthritis and chronic lameness.

### Materials and methods

Seventeen adult horses with a diagnosis of osteoarthritis and single limb chronic lameness were recruited. The COP was recorded using a wireless pressure measuring system (TekScan®) with sensors taped to the hooves (either fore- or hind limb, depending on lameness location). The COPp coordinates were further processed. Procrustes analysis was performed to assess the variability of single strides COPp and average COPp among strides, gaits, and limbs by calculating Procrustes distances (D-values). A linear mixed-effects model was run to analyse D-values differences for lame and sound limbs. Additionally, average COPp D-values and COPp hoofprint shape indices were compared for lame and sound limbs with the Signed Rank Test.

### Results

At walk and trot the single-stride COPp D-values were significantly lower in lame than in sound limbs (marginal effects p<0.001). Analysis of the average COPp D-values confirmed that each hoof COPp is highly consistent with itself over subsequent trials but is different from the contralateral. COPp and hoofprint shape indices did not differ between sound and lame limbs. Footstrike and lift-off within the hoofprint showed that most horses had lateral footstrike and lift-off, independently of the lameness location.

**Funding:** The present study was part of a larger trial evaluating the efficacy of a novel analgesic treatment in horses, which was funded by a Spark SNSF grant (CRSK-3_190256, received by CS; https://www.snf.ch/de) and by a grant from the ANALGESIA Institute Foundation and the DOMES PHARMA Group (https://www.domespharma.com/en/home/ received by CS). The funders had no role in study design, data collection and analysis, decision to publish, or preparation of the manuscript.

**Competing interests:** The authors have declared that no competing interests exist.

## Conclusion

Our findings are in line with previous observations that COPp are highly repetitive and characteristic for each horse and limb. There seems to be a further decrease in COPp variability in the presence of a painful limb pathology.

## Introduction

Objective gait analysis has gained increasing importance in equine orthopaedics. Its advantage over subjective assessment has been widely underlined in the recent literature [1,2], and using accelerometers or video analysis has made kinematic evaluation easily available for lameness grading in equine practice [2]. Furthermore, the value of kinetic parameters, such as force distribution, to objectify gait has been repeatedly highlighted [3–10]. In this context, the hoof centre of pressure (COP) and the centre of pressure path (COPp) are still largely unexplored. COP is the single theoretical point summarising all forces transferred to the hoof during the hoof-ground interaction at a certain point in time. COPp is the line that the COP follows from footstrike to lift-off during the stance phase of a stride. In 1987 Seeherman et al. [3] suggested that studying force propagation might be useful in evaluating equine lameness and hoof balance. Besides the reported applications in farriery [11–13], their systematic evaluation could indeed provide novel insights for characterising gait abnormalities and appendicular painful disorders. The prerequisite is, however, stride-to-stride repeatability to ensure reliable data. Recently, it was observed that COPp is very characteristic and highly repeatable for individual horses and limbs, comparable to a fingerprint. In healthy, non-lame horses, typical pressure patterns were recognized and described using averaged COP data recorded over three strides [14]. To date, no study has addressed the course of COPp in a population of horses with altered weight bearing due to osteoarthritic pain. In addition, while most prior studies have merely considered the average COPp, assessing single stride variability could provide valuable additional information regarding the adaptation of locomotion in the presence of pain.

The overarching aim of the present study was to characterize the COP and COPp in horses affected by chronic musculoskeletal pain and lameness due to osteoarthritis. Specific aims were: 1) To assess and compare the degree of similarity of single-stride COPp for the sound and the lame limbs at walk and trot; 2) To determine the degree of similarity of average COPp between limbs and their consistency over subsequent trials; 3) To calculate shape indices for the average COPp at walk and trot, to compare them for lame and sound limbs and to evaluate whether they correlate with an objective index of lameness severity. Based on human literature suggesting that pathological states might affect COPp variability [15–17], it was hypothesized that in case of lameness associated with chronic osteoarthritis, COPp recorded in subsequent strides from the lame limb would have a higher degree of similarity than the ones recorded from the sound limb. Specifically, it was expected that 1) COPp would show a greater degree of stride-to-stride consistency in the lame limb than in the sound limb; 2) average COPp would be consistent within a limb for subsequent trials but different from the contralateral limb, and 3) COPp shape indices would differ between sound and lame limbs, and the difference would became increasingly evident with increasing lameness severity, as assessed both subjectively and objectively. This hypothesis was based on a publication by Lopez et al. [18], which recorded COPp in dogs and found a shorter, cranialized, and reduced mediolateral excursion COPp in the lame limb compared to the sound.

## Materials and methods

### Animals

Seventeen privately-owned adult horses (9 mares and 8 geldings) affected by osteoarthritis and showing signs of chronic musculoskeletal pain were recruited within the context of a longitudinal clinical trial investigating a novel analgesic therapy (not published yet). The horses had a median age of 17.2 years (range 8–29 years) and a median body weight of 575.9 kg (range 430–670 kg). Demographic data are reported in Table 1. The horses included in the trial remained in their owners' care and were housed and fed in their usual surroundings. The owners signed informed written consent, and permission to perform the study was granted by the committee for animal experiments of the Canton of Bern, Switzerland, (Permission number: 32026).

**Table 1. Demographic data of the 17 horses affected by osteoarthritis and included in the trial.**

| Horse | Breed | Age | Sex | Weight (kg) | Current activity of the horse | Main affected limb | Main affected joint | Unshod/shod | History suggestive of bilateral disease | Grade of subjective lameness | MnD; MxD |
|---|---|---|---|---|---|---|---|---|---|---|---|
| H1 | Swiss Warmblood | 20 | mare | 600 | retired | right hind | pastern | unshod | no | 7 | 49R; 54R |
| H2 | Swiss Warmblood | 17 | mare | 670 | retired | left front | pastern | unshod | no | 7 | 129L; 3L |
| H3 | Freiberger | 12 | mare | 640 | retired | right hind | stifle | unshod | no | 6 | 13R; 30R |
| H4 | Swiss Warmblood | 18 | mare | 640 | retired | right front | coffin | shod | yes | 4 | 20R; 9R |
| H5 | Swiss Warmblood | 14 | gelding | 600 | retired | right front | coffin | shod | no | 7 | 92R; 35R |
| H6 | American Quarter Horse | 8 | gelding | 530 | retired | right hind | stifle | unshod | no | 5 | 15R; 25R |
| H7 | Holsteiner | 17 | gelding | 650 | pleasure riding | right front | pastern | shod only on the front limbs | no | 3 | 15R; 9R |
| H8 | Selle Français | 13 | mare | 670 | pleasure riding | left front | coffin | shod | yes | 4 | 21L; 16L |
| H9 | American Quarter Horse | 10 | mare | 530 | retired | left front | carpus | shod only on the front limbs | yes | 6 | 36L; 42L |
| H10 | American Paint Horse | 17 | mare | 430 | retired | right front | carpus | unshod | yes | 6 | 19R; 23R |
| H11 | Freiberger | 16 | mare | 540 | leisure | right front | fetlock | shod only on the front limbs | no | 4 | 32R; 4L |
| H12 | Irish Cob | 23 | gelding | 490 | pleasure riding | left front | fetlock | shod | no | 5 | 32L; 5L |
| H13 | Dutch Warmblood | 29 | gelding | 570 | pleasure riding | left front | pastern | shod | yes | 7 | 57L; 18L |
| H14 | American Paint Horse | 21 | gelding | 540 | pleasure riding | right hind | hock | shod | yes | 6 | 6R; 16R |
| H15 | Irish Warmblood | 16 | gelding | 640 | retired | right front | pastern | shod only on the front limbs | yes | 6 | 37R; 2L |
| H16 | Swiss Warmblood | 18 | mare | 500 | unknown | left front | pastern | unshod | yes | 6 | 29L, 23L |
| H17 | Unknown | 23 | gelding | 550 | pleasure riding | right front | coffin | shod | yes | 5 | 65R, 23R |

Lameness was scored using a scale ranging from 0 to 10 (0 = clinically sound, 10 = non-weight bearing lameness), based on Wyn-Jones et al. [19]. MnD (Minimal difference = difference between the vertical minima); MxD (Maximal difference = differences between the vertical maxima); based on Starke et al. [20]. R = right, L = left. For the frontlimb lameness the values derived from the poll were considered, for the hindlimb lameness the values from the pelvis were considered.

## Inclusion and exclusion criteria

An active recruitment campaign to find horses with a pre-existing diagnosis of appendicular osteoarthritis was conducted between March and June 2020. The inclusion criteria were age >3 years, history of single limb lameness of at least 3 months, presence of other typical clinical signs of osteoarthritis (such as abnormal gait, difficulty lying down or standing up, reduced performance or intolerance to being ridden) and radiological or MRI confirmation of osteoarthritis (at least one appendicular joint affected). Exclusion criteria were ongoing therapy with NSAIDs (in the 2 weeks prior), corticosteroids (in the 4 weeks prior) and the presence of systemic diseases and/or neurological disorders. A complete clinical and neurological examination was performed to confirm the absence of other clinical abnormalities. Seventy-two horse owners responded and applied to participate in the study. Twenty horses were initially considered eligible and were invited for a complete clinical examination. Finally, 17 horses met the inclusion criteria and were included in the study. Detailed information is provided in Table 1.

## Lameness scoring and gait symmetry analysis

Data considered in the present investigation were collected by a certified equine veterinarian (SA) with several years of experience. Lameness was evaluated and documented using subjective and objective measures. Based on the UK lameness score [19,21], lameness was subjectively scored using a scale of 0 to 10 (0 = clinically sound, 10 = non-weight bearing lameness) after observing the horses at walk and trot on a straight line (solid surface) as well as on a circle at walk. Thereafter, a validated system consisting of inertial measurement units (Xsens Technologies B.V., Enschede, The Netherlands) and custom-written software (Equigait Ltd, Cheshunt, Herts, UK) was used to quantify gait asymmetry objectively. Data was recorded and processed as previously described [20,22,23]. Lameness was defined as MnD (difference between the vertical minimal reached during left and right stance) and/or MxD (difference between the vertical maxima reached during left and right stance) greater than 6 mm for the forelimb and 3 mm for the hindlimb, based on thresholds established in prior studies [24,25]. Furthermore, the actual values of MnD and MxD, provided in the Equigait report, were used to quantify the extent of lameness as previously suggested [25]. Patient records were studied to identify histories suggestive of bilateral disease, as noted in Table 1. Based on the evaluation of parameters derived by the inertial sensors system, complex lameness cases were ruled out for all seventeen horses included in the study.

## Hoof pressure measurements

The hoof COP was recorded using a wireless pressure measuring system (TekScan Hoof System®, TekScan Inc, South Boston MA). Thin sensors with a spatial resolution of 3.9 sensels/$cm^2$ and pressure range of 0–200 N/$cm^2$ were cut to shape and taped to the hooves to record the pressure across the solar surface during the stance phase of locomotion. The limb subjectively and objectively showing the most prominent lameness (defined as a lame limb) and the contralateral limb (defined as a sound limb) were instrumented. The two sensors were connected to a data logger fixed on a girth behind the withers. Before each recording session, sensors were conditioned and calibrated using the walk calibration procedure as described by the manufacturer. For the purpose of the current study, uncalibrated (raw) force data were used. Data were sampled at 250 Hz and transmitted to a laptop. The research software provided by Tekscan (HOOF research Version 7.55–02, Tekscan Hoof System®, TekScan Inc, South Boston MA) allowed a real-time visualisation of the COP trace and offline analysis. Horses were guided at walk and trot on a straight line, with a loose rope by the owners on a hard surface over approximately 30 m targeting a constant speed. A minimum of ten to twelve strides of

steady-state gait were obtained for each run. Two trials at walk and one at trot were recorded for each horse.

## Data processing

ASCII files storing the coordinates of the COP over time for each individual stride were exported from the Tekscan acquisition software for all trials (walk and trot). The corresponding COPp was then reconstructed in Matlab (The Mathworks Inc, R2020a, Natick, MA, USA) for each stride and normalised to 501 points by linear interpolation. The normalization process was required to be able to proceed with further analysis as for most of the analysis a constant number of points is mandatory. Subsequently, an average COPp for each trial was calculated as an equally weighted mean of the individual strides COPp. For a direct comparison with the average COPp provided by the Tekscan software (which is specified by 51 points), an average COPp normalised to 51 points was also calculated in Matlab. Nevertheless, the decision was taken to continue with the average values calculated in Matlab, instead of the average calculated and provided by the Tekscan software, to ensure that the origin of the data is traceable.

A mathematical approach was used to compare the COPp for different horses and hoof print sizes; the average COPp was projected on the average hoof print given by the Tekscan software. The average hoof print displayed for each walk or trot trial was first imported into Matlab. An ellipse that best fitted the hoof print was computed using a purpose-specific Matlab script [26]. The centre of the ellipse was considered the origin of a coordinate system. In a second step, the average COPp data were coupled to the hoof print data by converting sensel units to centimetres, thereby allowing a precise superimposition of the different data on the coordinates system. This standard approach allowed reliable visualisation of the average COPp in relation to the hoof print, comparison between limbs, trials and horses and calculation of COPp shape indices.

## Data analysis

**Procrustes analysis.**   A Procrustes analysis was performed to compare the shape of the single-stride COPp within a limb in a trial (either walk or trot). The number of COPp coordinates was normalised to 101 points by linear interpolation, this number was chosen based on Nauwelaerts et al. [14] and for this reason differed from the number to which the points were normalized to for the average calculation. The Procrustes distances D between pairs of COPp were subsequently calculated using the built-in Matlab function *Procrustes*. The Procrustes distance measures the similarity between two shapes and assumes values ranging from 0 to 1, where 0 corresponds to identical shapes. To avoid measurement errors, the Procrustes analysis was always performed between COPp recorded during 2 strides. All the strides were compared in pairs in all possible combinations and always in both directions (i.e., stride 1 with stride 2 and then stride 2 with stride 1). The two analyses provided identical results.

With a similar approach, the degree of similarity of the average COPp for paired limbs (sound and lame) at walk and for the same limb during two consecutive walk trials was assessed using Procrustes and the evaluation of D-Values. This analysis was always performed using the average COPp previously calculated in Matlab.

**Shape index analysis for the averaged COPp.**   Following previously reported COPp shape indices in dogs [18] several indices were analysed on the standard hoof prints generated in Matlab and their average COPp. Details on how these indices were derived are shown in Fig 1 and Table 2.

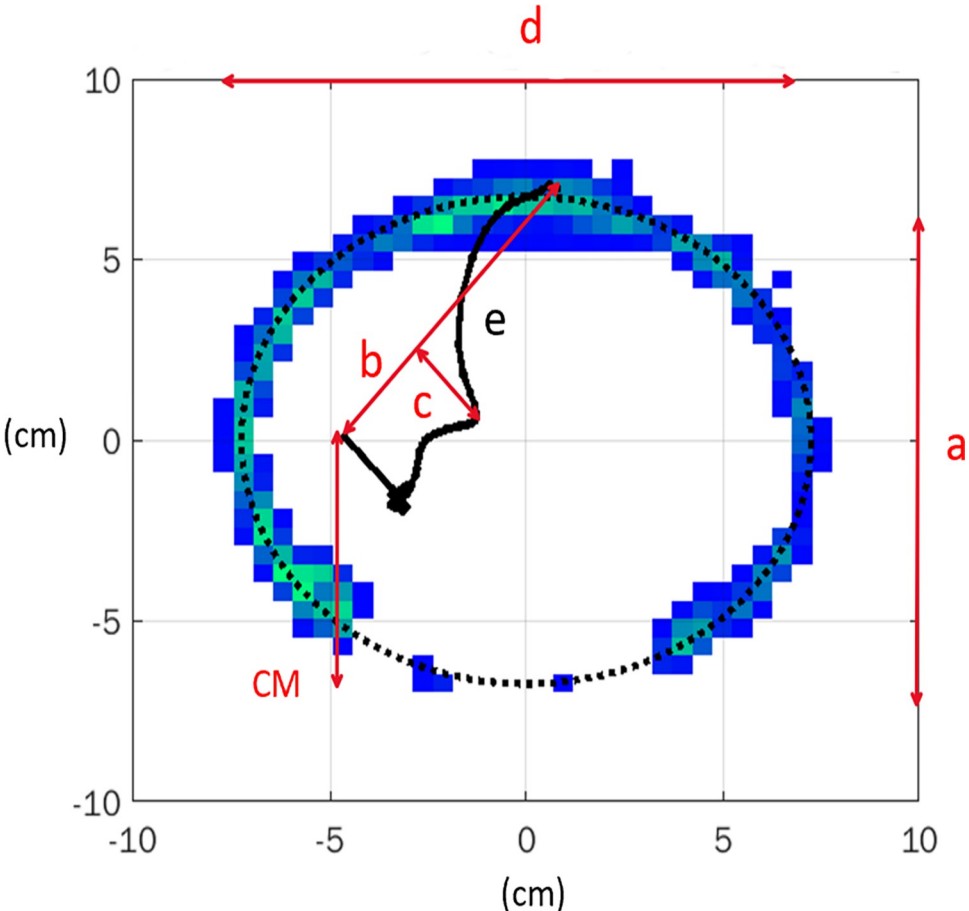

**Fig 1. Example of an ellipse fitted to the hoof print with the corresponding COPp.** See Table 2 for the definition of segments and indices.

**Footstrike and lift-off coordinates of the average COPp.** In addition to the previously described shape indices, footstrike and lift-off coordinates were also evaluated for the trot trials. Footstrike and lift-off were defined as first and last COP points on the average COPp. This analysis was only performed on trot trials to allow comparison with previous studies. The hoof prints were first fitted with an ellipse, as described above, and plotted together with their average COPp. Subsequently, each hoof print was divided into 4 quadrants, as illustrated in Fig 2. For left limbs, the quadrants were numbered from 1 to 4 in a clockwise direction, starting from the top left quadrant. For right limbs, the quadrants were mirrored and therefore numbered from 1 to 4 in a counterclockwise direction starting from the top right quadrant. This allowed a comparison of the limbs in the medio/lateral and plantaro-palmaro/dorsal plane.

## Statistical analysis

The Shapiro-Wilk test was applied to test the data for normal distribution. Due to the obtained results, data are presented as median and interquartile ranges (IQR) throughout the manuscript. For exploratory purposes, all the D-values calculated for the 17 horses were pooled together and graphically compared for lame and sound limbs at each gait (walk and trot). Then, a linear mixed effect model was run to compare D-values for lame and sound limbs, including horse as a random intercept and limbs and repetitions (corresponding to the first 20

**Table 2. Definition of standard distances and shape indices within the hoof print.**

| | |
|---|---|
| **a** | Vertical semi-axis of the ellipse fitted to the hoof print |
| **b** | Distance from the first to the last COPp point |
| **c** | Maximum distance of the COPp to the segment connecting the first and the last COPp coordinates |
| **d** | Horizontal semi-axis of the ellipse |
| **e** | Length of the COPp |
| **Caudal margin (CM)** | Distance between the Y-coordinate of the first COPp point and the Y-value of the vertex at the bottom point of the ellipse |
| **Craniocaudal index (CrCl)** | COP length (b) in relation to the hoof length (a), expressed as a percentage and determined using the following formula: % = (b / a) × 100 |
| **Centre of the pressure excursion index (CPEI)** | Latero-medial excursion of the COP (c) in relation to the hoof width (d), expressed as a percentage and determined using the following formula: % = (c / d) × 100 |
| **(CM/a) x100** | Caudal margin divided by total hoof length (a), expressed as a percentage |
| **(e/a) x100** | COPp length (e) divided by the total length of the hoof (a), expressed as a percentage |

The letters "a-e" are illustrated in Fig 1. The indices CM and CrCl, have been previously described for dogs [18]. The last 2 parameters are calculated to allow comparison among individuals.

D-values calculated per horse, limb, and gait) as fixed effects. The model was completed by adding potential presence of bilateral lameness and lameness location (front or hind) as covariates. The difference between sound and lame was investigated analysing the marginal effects for limbs (sound or lame) and repetitions, and the interaction term (limb*repetition). For each gait (walk and trot) a separate linear mixed model was run. Due to lack of normality distribution of the residuals, the two models were run again after Box-Cox transformation of the D-values. The linearity of both models was confirmed by a special post-hoc analysis ($x^2$ tests of linear hypotheses).

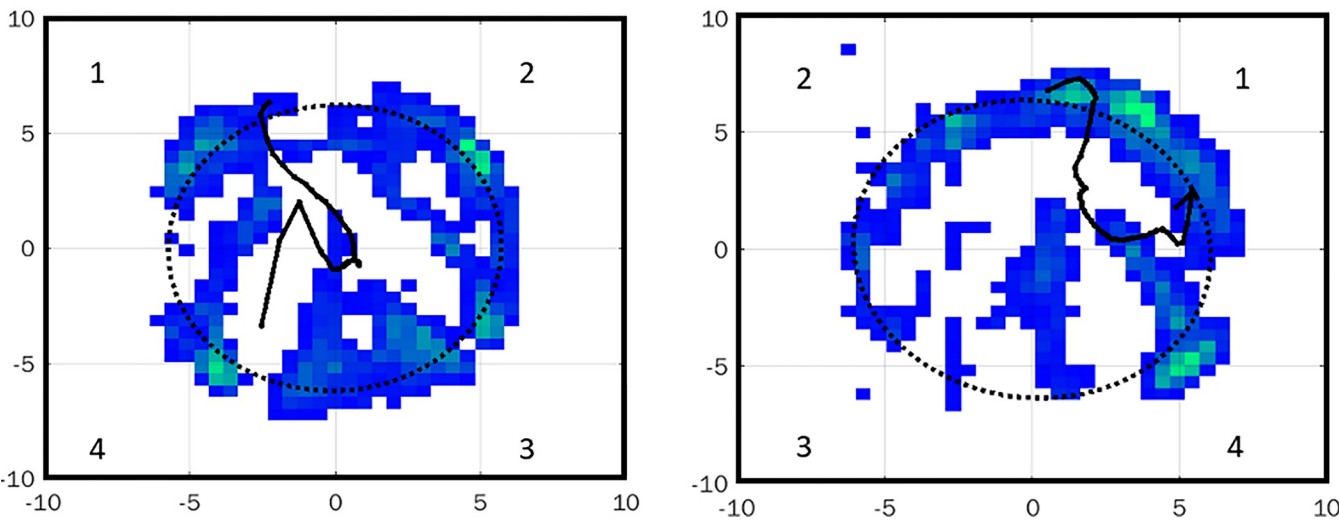

**Fig 2. Representative images of hoof prints and COPp for a left limb (left panel) and a right limb (right panel).** Each hoof print was divided into 4 quadrants. The numbering of the quadrants is illustrated for both hoof prints.

Hoof prints and COPp shape indices determined for sound and lame limbs were compared using the Signed-Rank Test. Similarly, a Signed-Rank Test was used to analyse the D-values obtained by comparing the average COPp for sound and lame limb at walk and the average COPp for the same limb during two consecutive walk trials.

Footstrike and lift-off coordinates of average COPp in relation to the quadrants were presented in a frequency table.

Statistical significance was set at 5% ($p < 0.05$). Statistical analysis was performed using Systat Software, SigmaPlot (Version 14) and STATA18 (StataCorp., College Station, TX, USA).

## Results

For each horse, trial, and limb, single COPp traces were reconstructed in Matlab and represented graphically (Fig 3).

The pooled D-values for lame and sound limbs at each gait (walk and trot) are shown in Fig 4 (Fig 4).

The linear mixed effect model revealed that, for walk and trot, the D-values were statistically significantly different for limb (sound versus lame) and repetition respectively (marginal effects for walk: 0.15, SE: 0.03, $p < 0.001$; marginal effects for trot: 0.18, SE: 0.03, $p < 0.001$), while for the interaction term, they were not statistically significantly different (p-value for limb*repetition: p = 0.361 for walk, p = 0.971 for trot).

Analysis of the average COPp D-values confirmed that each hoof COPp is consistent with itself over subsequent trials but differs from the contralateral limb.

Indeed, significantly lower D values (p<0.001) were documented for average COPp recorded from the same limb in 2 subsequent trials at walk [median 0.03 (0.01–0.08) for the lame limb; median 0.06 (0.03–0.08) for sound limb] than for average COPp recorded from the two limbs within the same trial [median 0.16 (0.08–0.2) for both walk trials].

### Shape index analysis

Descriptive statistics for the shape indices are reported in Table 3. No statistically significant difference was found for any hoof print and COPp shape index between lame and sound limbs.

The results of the analysis of the footstrike and lift-off coordinates of the average COPp are shown in Table 4. For the left limb, 10/17 horses had a latero-palmar/plantar footstrike, i.e. a

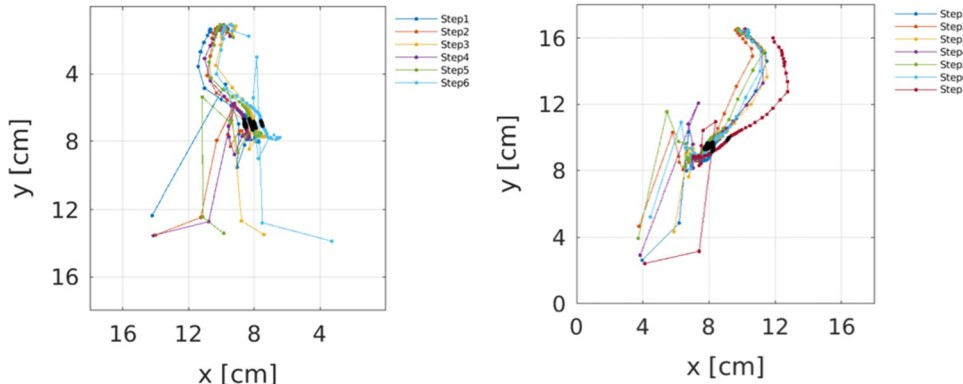

**Fig 3. Representative images of single COPp traces acquired during a trot trial from a horse lame on the right forelimb.** Left panel: Left hoof, right panel: Right hoof.

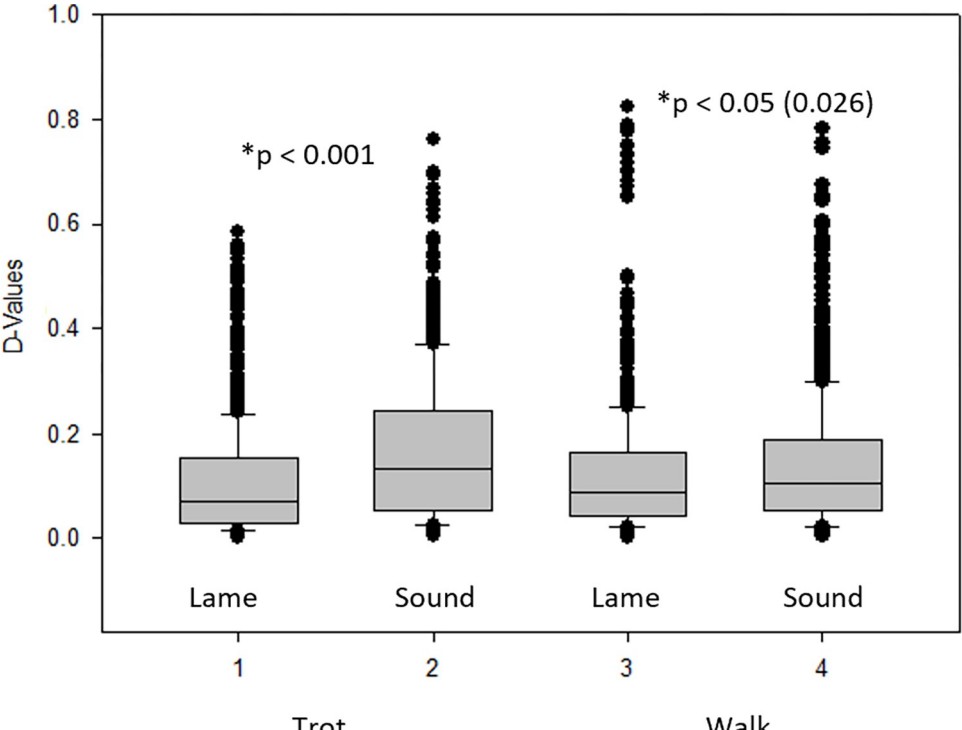

**Fig 4. Pooled D-values for trot (left) and walk (right) trials calculated for the 17 horses included in the study.** At walk, median D-values were 0.106 (0.053–0.190) and 0.089 (0.0436–0.165) for sound and lame limbs respectively, while at trot they were 0.132 (0.052–0.245) and 0.0723 (0.0301–0.154) for sound and lame limbs respectively.

footstrike in quadrant 4; 13/17 horses had a latero-dorsal lift-off, i.e. in quadrant 1. For the right limb, 7/17 horses had a latero-palmar/plantar footstrike and 15/17 latero-dorsal lift-off. In general, a higher prevalence of footstrike in quadrant 4 and lift-off in quadrant 1 was observed independently from the lameness location.

**Table 3. Results of the shape index analysis are shown.**

| Index | Limb | WALK1 (Median (25%;75%)) | P-Value | WALK2 (Median (25%;75%)) | P-Value | TROT1 (Median (25%;75%)) | P-Value |
|---|---|---|---|---|---|---|---|
| CM | L | 4.4 (2.4;7.9) | 0.431 | 3.9(2;7) | 0.611 | 4 (2.4;8.2) | 0.854 |
|  | S | 3.2 (1.5;7.8) |  | 3.5 (2.6;6) |  | 4.5 (2.2;6.7) |  |
| COP length e | L | 15.5 (13.6;17) | 0.132 | 15.4 (12.9;16.7) | 0.97 | 16.1 (13.8;17.4) | 0.678 |
|  | S | 16.6 (14; 18) |  | 15.6 (12.7;18.7) |  | 15.4 (12.7;18.3) |  |
| CrCL | L | 81.4 (61.2;89.6) | 0.353 | 83.3 (65.2;97.8) | 0.548 | 81 (54.7;92.8) | 0.854 |
|  | S | 84.5 (62.3;97.8) |  | 85 (63.8;96.3) |  | 74 (50.4;89.3) |  |
| CPEI | L | 12.5 (7;14.5) | 0.378 | 10.7 (6.7;13.3) | 0.243 | 13.5 (8.1;21.2) | 0.263 |
|  | S | 11 (8.3;17) |  | 11.7 (6.7;16.1) |  | 18 (10.8;22.6) |  |
| (CM/a) x100 | L | 33.4 (19.9;59.3) | 0.329 | 26.9 (15.4;52.3) | 0.89 | 30.5 (19.3;63.1) | 0.963 |
|  | S | 22 (12.1;58.5) |  | 23.6 (15.9;49.9) |  | 36.5 (15.757) |  |
| (e/a) x100 | L | 101.2 (101;101.3) | 0.12 | 101.2 (101.0;101.3) | 0.517 | 101.3 (101.1;101.3) | 0.404 |
|  | S | 101.3 (101;101.4) |  | 101.1 (101;101.3) |  | 101.3 (101; 101.3) |  |

From left to right: Shape indices from lame and sound limbs deriving from the first walk trial (WALK1), from the second walk trial (WALK2) and from the trot trial (TROT). Median and IQR values are reported.

**Table 4. Distribution for each quadrant (as in Fig 2) of the footstrike and lift-off coordinates of the average COPp for the left-right comparison and for the lame-sound comparison (Q = Quadrant).**

| Quadrant | Left vs. right comparison | | | | Lame vs. sound comparison | | | |
|---|---|---|---|---|---|---|---|---|
| | Q footstrike left | Q lift-off left | Q footstrike right | Q lift-off right | Q footstrike lame | Q lift-off lame | Q footstrike sound | Q lift-off sound |
| Q1 | 3 | 13 | 4 | 15 | 3 | 15 | 4 | 13 |
| Q2 | 1 | 4 | 3 | 1 | 3 | 1 | 1 | 4 |
| Q3 | 3 | 0 | 3 | 0 | 3 | 0 | 3 | 0 |
| Q4 | 10 | 0 | 7 | 1 | 8 | 1 | 9 | 0 |

Total amount of horses included 17 horses (N = 17).

## Discussion

The overarching aim of this study was to assess the characteristics and repeatability of COPp in horses with chronic osteoarthritic pain affecting either a fore- or a hindlimb. Previous evidence in horses suggested that COPp is a distinguishing characteristic of an individual equine hoof, similar to a human fingerprint [14]. The extent to which the COPp shape and repeatability can be affected by chronic pain and lameness had not been previously investigated.

Initial visual analysis of our recordings confirmed that the repeatability of the individual strides COPp within a walk or trot trial is high, as previously noticed [14]. Not only within a trial but also between trials and gaits (walk and trot), it seems that the basic shape of an individual hoof COPp does not change in most cases.

The first specific aim of the present study was to assess and compare the degree of similarity of single-stride COPp for the sound and the lame limbs at walk and trot. Based on human literature, it was hypothesised that in the case of lameness associated with chronic osteoarthritis, COPp recorded in subsequent strides from the lame limb would have a higher degree of similarity than the ones recorded from the sound limb. Indeed, it had been previously shown that the variability of COPp decreases due to a gait abnormality; this has been proven for human patients affected by hip instability and following stroke attacks [27–29]. With this in mind, we hypothesised similar findings in our study population of horses affected by chronic osteoarthritis pain. To assess the degree of similarity among single-strides COPp we calculated D-values using the Procrustes analysis as previously proposed [14]. Low D-values (close to 0) indicate that the similarity among shapes is high, while high D-values (close to 1) indicate high variability among shapes. The linear mixed effects model analysis showed that, as hypothesised, D-values were significantly higher for sound limbs than lame limbs, reflecting a higher similarity among COPp for the pain-affected limb than the contralateral. Confounding factors such as variable lameness location (front or hind) and possible presence of bilateral joint pathology were accounted for in the model. This allowed to increase results robustness despite the inhomogeneity of the samples included. Nevertheless, a potential impact of painful processes affecting other limbs on the COPp of the mainly affected limb cannot be entirely ruled out. Furthermore, it is important to consider that in the horses included in the present trial, pain was not specifically characterised (for example by use of quantitative sensory testing or in-depth behavioural analysis). Thus, we cannot make any assumption regarding peculiar pain characteristics and quality and its relation to the presented findings, considering that osteoarthritis-induced pain is complex and highly variable among affected individuals.

The significant effect of repetition found for the D-values is intuitively challenging to interpret. As the first twenty calculated D-values were considered in the mixed model as repeated measures, this finding could indicate that there might be a gait adaptation over time, as for both lame and sound limbs, at walk and at trot, the D-values became lower over the repetition

series. Even if the temporal relation is not direct for this parameter, first values represent the comparison of COPp recorded in the first strides of the trial, while later values are rather comparing later strides. Further work is surely needed to understand if this observed D-values behaviour has a true biological meaning or not.

Our second specific aim was to determine the degree of similarity of average COPp between limbs and their consistency within limbs over subsequent trials. Averaging COPs allowed a direct comparison of COPp consistency over time and for individual hooves, as well as a comparison of predefined shape indices following a geometrical assessment. As for single-strides COPp, similarity among average COPp shapes was tested using Procrustes analysis and D-values. Our findings align with those described by Nauwelaerts et al. [14] who described high repeatability of average COPp shape for individual hooves between trials but not necessarily between hooves in the same horse and trial. At this stage, it is difficult to rule out if this difference between the two hooves within one trial is coming from the fact that the COPp is a priori different between limbs or if the chronicity of the process affecting one of the limbs has an influence. In order to better understand this phenomenon, it would be necessary to perform a similar study on healthy limbs before and after the appearance of a painful process to see if the chronicity of a painful process affects the COPp. In contrast to the three strides used for comparison in the previous study, the equipment used in the present study allowed the continuous recording of a large number of COPp. Indeed, the introduction of wireless pressure sensors to be applied under the hoof has made it possible to easily acquire force distribution data for multiple strides in field settings [11,30–32]. We confirm that within a trial, between trials and between gaits (walk and trot), the basic shape of an individual hoof COPp remains the same.

Our third specific aim was calculating shape indices for the average COPp at walk and trot and comparing them for lame and sound limbs. In dogs affected by elbow dysplasia and lameness, shape indices differed between the sound and lame limbs. In particular, the COPp started further cranially under the paw of the affected limb and was shorter in general when compared to the COP of the sound limb [18]. It can be assumed that this phenomenon results from learning to load the limb in a fashion that minimises pain consistently. We hypothesised that selected shape indices would differ for sound and lame limbs in horses as well. Contrary to what was described for dogs, we could not find any significant difference between lame and sound limbs for any of the COPp shape indices. It has been previously pointed out that not all species have the same potential to adapt footing patterns to avoid a painful situation. The weight and size of the animal appear to dictate the degree to which this adaptation is applied [33]. In addition, the rigid nature of the equine hoof (in comparison to a canine paw) surely further reduces the potential to adapt footing in a way that affects COPp characteristics significantly.

In addition to the shape indices, the footstrike and the lift-off coordinates of the average COPp were evaluated for the 17 horses. This allowed a direct comparison with previously published data [14]. In the horses of the present study, a majority of footstrike and lift-off occurred on the lateral side of the hoof. This finding agreed only in part with previous observations [14] in which most horses had a latero-dorsal footstrike. A former study [34] observed a tendency towards a palmar landing with a continuation of the COPp in a dorsal direction, which agrees with our results. Both previous studies were conducted on healthy, non-lame horses. Based on our findings in lame horses, it does not seem that joint pain and lameness significantly affect the footstrike and lift-off patterns as they are similar in sound and lame limbs and in line with those previously described in healthy horses.

Since COPp appear to be very repeatable within a trial and over time, one can assume that even small changes in the COPp may be interpreted as an altered force distribution. With serial measurements, COPp may have clinical potential in recognising pathological gait

changes or verifying the positive effects of an intervention. Several previous investigations reported the effects of farriery on COPp [11–13]. Gross visual evaluation of the COPp within the hoof print was proposed to assess the effect of hoof load distribution in shod versus unshod horses, the effectiveness of corrective farriery, and the influence of several ground properties and shoeing intervals [11–13]. Based on the methodology we established and on our preliminary findings, future studies might focus on the influence of hoof shape and shoeing on lameness and COPp variability, thus contributing to advance the field of equine podiatry.

## Limitations of the study

This study focused on data collected from horses affected by chronic lameness and osteoarthritis.

The lack of a control population is the main limitation in this study. While COPp from healthy, non-lame horses were described in a previous work, they cannot be directly compared with those of the present study, as the recording methods used were different. Thus, it would be ideal to apply the same methodology to a cohort of healthy and sound horses in future studies. This would reliably confirm or reject the hypothesis that higher variability of COPp accompanies normal gait and that perhaps decreased variability might be used as an early indicator of painful processes affecting locomotion. With regard to lameness location, all horses were enrolled in the study following prior localisation of lameness by the private veterinarian (including regional or intraarticular diagnostic analgesia) who diagnosed osteoarthritis in the first instance. Based on the sum of findings evaluated for inclusion in the present study (history, clinical examination, lameness examination and radiographic findings), we judged that repeating diagnostic analgesia was not necessary, but we acknowledge that this is a limitation. Even though all the horses included in this study showed obvious single limb lameness (making it the reference lame or most lame limb), we cannot exclude the presence of bilateral disease or multi-limb lameness. Nevertheless, according to recent evidence [25], closer evaluation of the kinematic data and the inclusion of a withers sensor, allowed us to rule out multi-limb lameness (in the sense of front-end influencing hind-end and vice versa) as a criticism [25]. Furthermore, we recruited a rather inhomogeneous population of horses, showing lameness that could interest either the front or the hind limb. Focusing on one single location and possibly on a single joint would have increased comparability among individuals, but temporal and financial constraints did not allow to follow this ideal approach. Finally, the presence or absence of shoes and the shoeing stage at the time of recordings were not incorporated into our interpretation of COPp data. Similarly, shod and unshod horses were included in the study by Nauwelaerts et al. [14]. While the influence of shoeing details should certainly be considered in future studies, published evidence seems to attribute minimal influence of shoeing stage on COPp [12,35]. Therefore, shoeing state is not expected to modify COPp variability among strides, especially when the COPp are derived from limbs of the same horse with the same shoeing status.

Another limitation is that the technology applied to measure COPp in the present trial has not been validated against a gold standard in veterinary medicine, yet it has been used in various clinical trials [11,32,35,36] and was previously validated in human medicine. Furthermore, the walk calibration performed cannot be considered adequate for accurate pressure measurements in the presence of gait asymmetry and different calibration methodologies would need to be applied if absolute pressure values would be targeted. For the purpose of the present study, uncalibrated, relative force data were used to describe the COP position during locomotion. They originate from the raw output voltage of the single measuring cells that constitute the sensors. To account for potential slight variability in the voltage output among sensing

elements, an equilibration procedure should be carried out to render the output uniform. This requires the use of a special device able to apply perfectly homogenous pressure on the whole sensing surface. As such a device was not available at the time of data recording, equilibration was not performed. While we acknowledge that this is a limitation, we believe that the consequences of missing cell equilibration can be considered negligible in the context of COPp characteristics analysis.

## Conclusion

Similar to the findings in Nauwelaerts et al. [14], the present study confirmed that the hoof COPp is highly characteristic for each horse and limb, similar to an individual fingerprint. The COP trajectory is very repetitive among strides and gaits for individual animals, both for the lame and sound limbs. Significantly lower D-values, indicating lower COPp variability, were found for the lame compared to the sound limb at both walk and trot, independently from lameness location (front versus hind) and potential presence of bilateral pathology. The analysis of shape indices and positioning of footstrike/lift-off did not reveal significant differences between sound and lame limbs. The proposed standardized methodology to reconstruct single strides COPp is now available for future studies that might target the evaluation of COPp in different contexts and over time.

## Supporting information

**S1 Table. Metadata D-values individual horses.**
(XLSX)

**S2 Table. Metadata mixed model.**
(XLSX)

**S3 Table. Metadata shape indices.**
(XLSX)

**S4 Table. Metadata D-values AvrageCOPp.**
(XLSX)

## Acknowledgments

We would like to thank the horse owners for their cooperation in the data acquisition phase and Marilu' Garo for her contribution to the statistical analysis.

## Author Contributions

**Conceptualization:** Larissa Irina Buser, Stefan Witte, Claudia Spadavecchia.

**Data curation:** Larissa Irina Buser, Nathan Torelli, Sabrina Andreis, Stefan Witte, Claudia Spadavecchia.

**Formal analysis:** Larissa Irina Buser, Stefan Witte, Claudia Spadavecchia.

**Funding acquisition:** Claudia Spadavecchia.

**Investigation:** Larissa Irina Buser, Sabrina Andreis, Stefan Witte, Claudia Spadavecchia.

**Methodology:** Larissa Irina Buser, Nathan Torelli, Claudia Spadavecchia.

**Project administration:** Larissa Irina Buser, Stefan Witte, Claudia Spadavecchia.

**Resources:** Claudia Spadavecchia.

**Software:** Nathan Torelli.

**Supervision:** Stefan Witte, Claudia Spadavecchia.

**Validation:** Larissa Irina Buser, Nathan Torelli.

**Visualization:** Larissa Irina Buser, Nathan Torelli.

**Writing – original draft:** Larissa Irina Buser, Stefan Witte, Claudia Spadavecchia.

**Writing – review & editing:** Larissa Irina Buser, Nathan Torelli, Sabrina Andreis, Stefan Witte, Claudia Spadavecchia.

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
