## [Decision Letter · Decision Letter 0]

5 Jun 2023

PONE-D-23-12780Evaluation of the hoof centre-of-pressure path in horses affected by chronic osteoarthritic painPLOS ONE

Dear Dr. Buser,

Thank you for submitting your manuscript to PLOS ONE. After careful consideration, we feel that it has merit but does not fully meet PLOS ONE’s publication criteria as it currently stands. Therefore, we invite you to submit a revised version of the manuscript that addresses the points raised during the review process.

Please review your article in light of the comments provided by the reviewers. I am particularly interested on your thoughts regarding diagnosis and absence of a control group. The reviewers have provided many suggestions and I am sure you will find them useful. 

We look forward to receiving your revised manuscript.

Kind regards,

Aliah Faisal Shaheen

Academic Editor

PLOS ONE

3. Please make sure that all information entered in the 'Ethics Statement' section regarding ethics approval is also included in the Methods section of the manuscript

“The present study was part of a larger trial evaluating the efficacy of a novel analgesic treatment in horses, which was funded by a Spark SNSF grant (CRSK-3_190256, received by CS; https://www.snf.ch/de) and by a grant from the ANALGESIA Institute Foundation and the DOMES PHARMA Group (https://www.domespharma.com/en/home/ received by CS).  “

Additional Editor Comments:

The reviewers have submitted thorough commentaries and suggestions for your article that I hope you find useful. Please provide a point-by-point response with any changes made to the manuscript.

Reviewers' comments:

Reviewer's Responses to Questions

**Comments to the Author**

1. Is the manuscript technically sound, and do the data support the conclusions?

Reviewer #1: Partly

Reviewer #2: Partly

2. Has the statistical analysis been performed appropriately and rigorously? 

Reviewer #1: I Don't Know

Reviewer #2: No

3. Have the authors made all data underlying the findings in their manuscript fully available?

Reviewer #1: Yes

Reviewer #2: Yes

4. Is the manuscript presented in an intelligible fashion and written in standard English?

Reviewer #1: Yes

Reviewer #2: Yes

5. Review Comments to the Author

Reviewer #1: The manuscript presents an interesting study on hoof centre of pressure path in chronically lame horses. The number of included horses is relatively large for this type of study, and the study group is well defined. A limitation is that no control group with healthy horses is included. Another limitation is that it was not confirmed that the included horses were only affected in one limb at the time of measurement, since no diagnostic algesia was performed to confirm the localization of pain. This needs to be considered especially given that several horses had a history suggestive of multi-limb lameness. Another limitation is that, to my knowledge and according to the information presented in the manuscript, the measurement equipment used has never been validated against any kind of gold standard, for example force plate data, which means that the level of measurement errors is unknown. Even with these limitations, the study is still interesting since there is limited data available on this topic. However, there are a number of unclarities regarding how the data were analyzed that need to be addressed, and it is possible that some parts of the statistics need to be redone, please see my comments below.

Page 8

Line 26. Unclear what this meant by objective lameness examination in this context.

Line 34. What is the meaning of the ending s in COPps? Plural?

Page 9

Line 57. Unclear what you mean by indices (which is also spelled incorrectly here).

Line 63.”To date; no study has addressed the course of COPp in a population of horses with altered weight bearing due to pain.” This is incorrect. You have missed you include the following study in your reference list:

Evaluation of an in-shoe pressure measurement system in horses 10.2460/ajvr.2001.62.23

Page 10

Line 71. “calculate shape indices for the average COPp at walk and trot” Again unclear what you mean by indices.

Page 12

Table 1. Note that the PLOS formatting guide instructs that the table legends should appear above and not below the table. This applies to all tables.

Line 107. “foreseen for the analgesia trial” Odd wording. Rephrase?

Line 123. Please provide details on the calibration procedures used for the sensors. With this type of pressure sensor, frequent calibration is crucial to obtaining correct results.

Page 13

Line 126. Please specify exactly how you determined which limb that was considered lame and sound, including how you combined the subjective and objective scores.

Line 128. Not sure what you mean by custom made. Did you write your own software for this purpose? Or was this proprietary software that came with the measurement equipment?

Line 132. Does this mean that a minimum of 20 strides total were collected for each horse in walk but only 10 trot strides? Why?

Line 137. Here you say 501 points but under data analysis it says 101 points. Am I missing something?

Line 138. Why did you want to make this comparison? Also, the results for this comparison are not reported anywhere.

Line 141. It’s not obvious why this normalization process was undertaken, given that normalization is inherent to the Procrustes algorithm. Was this for calculating the shape indices that you describe further below? Please specify for which analysis/analyses that these normalized data were used.

Page 14

Line 153. I interpret this that you are comparing strides that come from the same horse, hoof and gait. In that case, why was it relevant to use Procrustes for this? No scaling, translation or rotation should be necessary to compare the center of pressure paths in that case, unless the sensor position relative to the hoof was unstable. If that was not the case a simple root mean square difference should give you the same result. That is, you can apply the formula for calculating the D-value directly to the data without transforming it first. Please explain the rationale for using this analysis. Further, when comparing multiple measurements it is more appropriate to use generalized Procrustes analysis, which the Matlab function that you have used doesn’t do. Please explain why you chose this approach.

Find 155. What did you input as target shape and comparison shape to the matlab procrustes function, respectively? Note that D-values returned from this function are normalized to the target shape. Please report this both for comparing strides from the same trial/hoof, and for comparing sound and lame limbs. Also, since it is known that chronic lameness can influence hoof shape over time, is it possible that this has confounded the D-values when comparing sound and lame limbs?

Figure 1. The hoof print in this figure looks weirdly distorted, it seems that the width/height relationship is off. Also, please capitalize Cm in the same way you have done in table 2 and please indicate the unit of the X and Y axes.

Line 163. I assume this should be table 2 rather than table 3?

Page 15

Table 2. Isn’t Cm also illustrated in Figure 1?

Line 172. Please specify how footstrike and lift-off events were defined and determined.

Page 16

Line 189. “were initially analysed for all 17 horses together and subsequently as independent variables” I don’t understand this part of the sentence. From reading the next sentences I assume this means that you made a group-level analysis ignoring that each horse has two limbs, and then a pairwise analysis taking this fact into account. I don’t quite understand why the first type of analysis makes sense in this context though, see my following comment.

Line 190. The Mann Whitney U doesn’t take repeated measures into account. This means that this analysis can’t be used for this dataset since each horse contributes one value for the left and one value for the right limb (minimum, more if you don’t use a summary measure per limb). I suggest that you use a repeated measures ANOVA or a mixed model, that includes horse and limb nested in horse as random variables (assuming stride-by-stride data) and transform the outcome variable as needed to achieve normal distribution of the residuals.

Line 191. “To characterise the behaviour of the single-strides COPp and the resulting D-values in individual horses, a subject-based analysis was performed using the Wilcoxon Signed Rank test to compare limbs for each gait.” I’m not entirely sure what you have done here. From my perspective it makes the most sense to compare sound and lame limbs within each horse and across all horses, using a summary measure, e.g. the median D-value, for each horse and hoof. From the legend for figure 4 it seems that you have indeed done this. However, you have failed to describe what summary measure you used. From the results text, it seems that you have also applied this analysis on an individual level. That’s not exactly great since the sample size will depend on the number of strides recorded, and since consecutive strides hardly can be considered independent measures, which is an underlying assumption of the statistic.

Line 195. You might want to be consistent with how the signed rank test is capitalized or not. Compare this and the previous sentence.

Page 18

Tables 3 and 4. I don’t understand how this is two tables. It would make more sense to make them table 3 A and B if you want to have a separate annotation for each panel.

Line 224 cont. This statistical evaluation is not described in the statistics section. Please describe how this test was applied and what statistic that was used.

Page 19

Table 5. The table contains unexplained abbreviations. You may want to refer to table 3 for this. “From left to right: shape indices for lame and sound limbs deriving from the first walk trial (WALK1), from the second walk trial (WALK2) and from the trot trial (TROT).” Is there a specific reason why the two walk trials are reported separately in this table but not otherwise?

Supporting information. The Excel document contains one sheet with D-values, however, it isn’t specified to which horse each value belongs. Further, since there are a lot of missing values in the end of some columns but never further up, it seems that all values on the same row do not belong to the same horse. This means that these data can’t be used to reproduce your statistics. Please update this sheet to include horse information.

Reference list. A few of the references are in correctly formatted or like information, e.g. name of the journal, e.g. nr 3 and 28.

Reviewer #2: This paper seeks to look at COP in horses with chronic OA.

Title: if definitive localization of the lameness via nerve or joint blocks was not performed the title has to change to just indicate lameness and not lameness due to chronic oa.

General Abstract Comments:

ideally indicate the number each of fore and hind limb lamenesses and their sites.

Are these bilateral lamenesses (as chronic oa tends to be)? Aka the “sound” side may still be affected. How were they diagnosed as chronic OA?

I am a little concerned about lumping different lameness and sites and front and hind limbs together.

Line 42- is the lateral footstrike on all limbs or only the lame ones?

40-41- p values? Quantification of the difference? Statistical tests used?

44 by typical do you mean unique to each horse? Typical makes it sounds as if all horses have the same response, which the second part of that sentence doesn’t support. Highly repetitive within a horse correct?

INTRODUCTION

60. could you clarify this phrase? Is consistency the best word? What is this in relation to? Objective lameness evaluation? Only COP measurement? Do you have concerns about achieving this in lame horses?

66 would single stride data be more variable and not as representative as averaging several strides? How do you “pick” the stride? Potential bias from that process- please justify further

74- please add into the intro the literature from which you based your hypothesis that the COPp would be more consistent in lame horses.

77 difference between front and hind limbs? Sound and lame limbs? Limbs on different horses?

79 do you mean objective lameness severity? Positively or negatively correlate? Please clarify

METHODS

91- radiographic presence of OA does not necessarily correlate to clinical lameness. Were these horses blocked (nerve or IA) to prove that the OA was the source of the lameness vs. soft tissue origin? How was the site of lameness localized? Without this information, you can state you studied COP in lame legs, but I don’t think you can positively state that it was chronic OA related. This may require a title change subsequently.

95 many of these horses are retired, how can you then use inclusion criteria of decreased performance or intolerance to being ridden?

Table 1- subjective lameness grade should be referenced and explained. Was there a inclusion criteria for subjective lameness?

Table 1- I appreciate your indicating bilateral or not for affected limbs, how did you assess the issue with the other limb? Did you compare COP’s to the other potentially affected limb in these instance or? Aka what did you use for “sound” comparisons? How did you decide you could pool together front and hind limbs? GRF in front vs hind limbs are different and COPs likely will reflect this too. Is there a refernce which has pooled front and hind together before validating this? Who assessed the subjective lameness? One person? Many? Intraclass correlations for that if many? Can you describe how they were assessed? Footing? Circles? Etc? at the same farm or ? What are the qualifications of the vet/s who diagnosed them?

97- corticosteroids can have anti-inflammatory effects that last longer than 4 weeks- if pain returns, it is suggestive of a soft tissue component to the issue or a misdiagnosis of the joint that is painful, please discuss/explain

99 was blood used to look for other medical issues? Clarify please if this related to part of your exclusion of other medical issues indicated in the first part of the sentence. As it is written now it sounds as if it is something else and not for that reason.

100 17 horses out of how many? How were they located? Previous clients? Elicited via an email for the research?

106- qualifications? One vet only?

107 foreseen is not the correct word here, maybe just delete?

109 may be an aberrant | after lameness? I’d like to see this paragraph above the table as this explains the subjective lameness grades. I think there should be an exclusion on the high end of the lameness scale for this study, no?

119 Do you have references that you can cite that indicate that if bilateral lameness is present that the Xsens will always pick up on it in horses that aren’t blocked? According to Keegan et al. AJVR 2012 “By contrast, the inertial sensor system measures the asymmetry of head movement between the right and left portions of a stride. Horses with bilateral forelimb lameness, in which the severity of lameness in opposite limbs is equivalent in every stride, would be judged as symmetric if inertial sensors were used…but the inertial sensor system cannot measure the absolute severity of lameness in 1 limb in any 1 stride.” So that the inertial sensors can pick out the lamer of paired legs only. This needs to be further detailed/justified/supported if you are claiming that the opposite leg serves as a control based on xsens inertial sensor data.

Also how did, or did you address/recognize compensatory lameness?

119- I don’t know if this journal allows you to refer to items by brand names (equigait)

126 bilateral lameness, especially when there is evidence for it in the history, ideally would have had see the horse’s lamest leg blocked to show that the opposite leg didn’t become sore.

131 what did you do to target a constant speed? Did you have markers set up to measure acceleration or deceleration?

191-193 what did you do to correct for multiple comparisons?

204 have you looked at front alone and hind alone? I am leary of pooling front and hind together, or would like to see the numbers and stats that you have to demonstrate they are ok to be pooled.

214 please clarify significant and trend? Trend normally indicates a non significant finding so these appear contradictory.

Table 3 and 4, are these corrected p values in any way for the number of comparisons indicated? Also this is confusing as the p value looks to be just looking at within a horse between the lame and sound leg? But I think you compared just overall lame to sound? Are some of these p values repeated as WIlcoxin should take into account all lame vs all sound. Some of these p values may reflect type 1 errors and become non significant if adjusted for the number of tests.

Table 6 define Q, clarify division by left right, include N=17 describe how this related to lateral sidedness.

230 p values ofter just go to one or 2 digits beyone the . sometimes you are 2 and other times 3, please see journal conventions

DISCUSSION

246-250 this is too intro like, please address whether you accept, partially accept or reject your specific hypotheses briefly and concisely, then go into details comparing to lit in later paragraphs.

Paragraph 255 too intro and method like, need to compare your results to the literature as the focus.

269 decreases in variability… is higher…. Is unclear. Please clarify

277 phenotyped how? I’m still uncomfortable claiming the pain is definitively OA pain, need to mention different joints can express things differently in the limitations if not here more explicitly

278 at the individual what? I would like to see statistics behind this claim that it wouldn’t affect things. I particularly need to see data of front vs hind, to understand if they can legitimately be pooled.

280- lack of diagnostic analgesia to localize the lameness should be explicit earlier and in the limitations. Given this information, I think you have to rephrase the paper in light of lameness and not chronic oa. You can include in the chart areas of previous concern from oa, and should include historical data of the length of lameness in the leg (over the short and long term).

305-306 too method like, delete

308 this is somehow not surprising, is not clear please rephrase

315 I dislike tendency, if this was NOT statistically significant it should not be discussed as if it is, please remove and just say this wasn’t significantly clear in this study but previous studies found….

328 Nauwelarts didn’t observe this study please rephrase, just say Similar to the findings in X, the present study confirmed…

330 confirm if this is lame and sound legs or just one of these

331-332 remove this sentences as this wasn’t significant There seems to be a tendency for even

331 lower COPp variability in the presence of a painful limb pathology, possibly indicating a preferential

332 way of hoof loading to decrease discomfort.

337 limitations- does this journal request this to be last? Normally this is before conclusions.

341 lameness and pain free is awkward, please rephrase

346 -347 what references are these? Please cite them here and above

350 please indicate who was shod or not in the table

351 references for farriery? This seems to be appropriate to work into other aspects of the discussion

356-358 this sentence is unclear, please simplify

References

Journal names seem differently referenced, AJVR and others are totally spelled out while Equine Vet J is partially shortened as are others, please fix per journals standard

6. PLOS authors have the option to publish the peer review history of their article (what does this mean?). If published, this will include your full peer review and any attached files.

Reviewer #1: No

Reviewer #2: No

---

## [Author Response · Author response to Decision Letter 0]

19 Jul 2023

Answers to Reviewer 1

Reviewer #1: The manuscript presents an interesting study on hoof centre of pressure path in chronically lame horses. The number of included horses is relatively large for this type of study, and the study group is well defined. A limitation is that no control group with healthy horses is included. Another limitation is that it was not confirmed that the included horses were only affected in one limb at the time of measurement since no diagnostic algesia was performed to confirm the localization of pain. This needs to be considered especially given that several horses had a history suggestive of multi-limb lameness. Another limitation is that, to my knowledge and according to the information presented in the manuscript, the measurement equipment used has never been validated against any kind of gold standard, for example force plate data, which means that the level of measurement errors is unknown. Even with these limitations, the study is still interesting since there is limited data available on this topic. However, there are a number of unclarities regarding how the data were analyzed that need to be addressed, and it is possible that some parts of the statistics need to be redone, please see my comments below.

Dear Reviewer, thank you very much for your constructive comments and remarks. They will improve the quality of the manuscript and we have done our best to consider all of them. 

The lack of a control population is, as always, a large limitation in this study and under different circumstances would have been a valuable preliminary study, prior to studying abnormal horses. The data on abnormal horses we present stemmed from a different study looking at the effect of a novel analgesic. We would be interested in studying normal horses in the future. At the same time we felt that despite the limitations the majority of horses served as their own controls and so justified publishing this set of data. With regard to lameness location; all horses were enrolled in the study following prior localisation of lameness (including local analgesia). Based on the sum of findings evaluated for inclusion in the study (history, clinical examination, lameness examination and radiographic findings) we decided that repeat localisation of the site of pain was not neecessary. This is a limitation. Potential bilateral limb lameness as a limitation to our study has been included in the limitations section. The majority of horses were obviously worse on either the left or the right limb (making it the reference lame or most lame limb). Closer evaluation of the kinematic data (and the inclusion of a whithers sensor) excludes multi-limb lameness (in the sense of front-end influencing hind-end and vice versa) as a criticism. Kinematic evaluation through use of the equipment and sensors used in this study has been validated against gold standard force plates and has been reported on a number of occasions (for example; Pfau T, Witte TH, Wilson AM (2005): A method for deriving displacement data during cyclical movement using an inertial sensor. J Exp Biol 208(Pt 13): 2503-2514 and Warner SM, Koch TO, Pfau T (2010): Inertial sensors for assessment of back movement in horses during locomotion over ground. Equine Vet J Suppl (38): 417-424).Validation of the Tekscan hoof pressure measurement method is limited in the veterinary field. Reports that include validation in human medicine include: Wirz D, Becker R, Li SF, Friederich NF, Müller W. Die Validierung des Tekscan-Systems für statische und dynamische Druckmessungen am humanen Femorotibialgelenk [Validation of the Tekscan system for statistic and dynamic pressure measurements of the human femorotibial joint]. Biomed Tech (Berl). 2002 Jul-Aug;47(7-8):195-201. German. doi: 10.1515/bmte.2002.47.7-8.195. PMID: 12201014 ).

Page 8

Line 26. Unclear what this meant by objective lameness examination in this context.

We have modified the text and removed the word objective.

Line 34. What is the meaning of the ending s in COPps? Plural?

Yes, the s was added in the case of ‘center of pressure paths’ as the plural. Since this may cause confusion, COPp is now used in the singular and the plural throughout the text. The manuscript has been altered accordingly. 

Page 9

Line 57. Unclear what you mean by indices (which is also spelled incorrectly here).

The text has been altered so that it now reads ‘ In 1987 Seeherman et al. suggested that studying force propagation might be useful in evaluating equine lameness and hoof balance’. This more accurately describes that author’s contribution.

Line 63.”To date; no study has addressed the course of COPp in a population of horses with altered weight bearing due to pain.” This is incorrect. You have missed you include the following study in your reference list:

Evaluation of an in-shoe pressure measurement system in horses 10.2460/ajvr.2001.62.23

While the study you mentioned evaluated the Tekscan technology, it does not study COPp. It has been included in the limitation section in which we discuss studies in the veterinary field that have used Tekscan technology for clinical studies (albeit without validation against a gold standard). Also included are studies by Judy CE, Hagen and Fürst. 

Page 10

Line 71. “calculate shape indices for the average COPp at walk and trot” Again unclear what you mean by indices.

In this instance we wanted to draw a parallel with the publication in small animals by Lopez et al. This publication uses the term indices to describe various parameters that they extract from the COPp for comparison between dogs. To enable direct comparison, we also used the terms index and indices. Perhaps parameter would have been a better choice? The indices are however precisely defined in Table 2.

Page 12

Table 1. Note that the PLOS formatting guide instructs that the table legends should appear above and not below the table. This applies to all tables. 

Thanks for this. We have corrected the locations of the table legends.

Line 107. “foreseen for the analgesia trial” Odd wording. Rephrase?

The sentence has been shortened and now only reads as follows; ‘Data considered in the present investigation were collected by an experienced equine veterinarian’. 

Line 123. Please provide details on the calibration procedures used for the sensors. With this type of pressure sensor, frequent calibration is crucial to obtaining correct results.

The COP data presented in the manuscript are merely describing the spatial distribution of pressure during stance, not pressure itself. Therefore, no calibration was necessary and raw data output from the sensors was sufficient to reconstruct the COP path.

Page 13

Line 126. Please specify exactly how you determined which limb that was considered lame and sound, including how you combined the subjective and objective scores.

All horses presented with a history of showing lameness in a single limb. The sum of both the subjective and objective lameness evaluation confirmed the allocation of lame and sound limbs.

Line 128. Not sure what you mean by custom made. Did you write your own software for this purpose? Or was this proprietary software that came with the measurement equipment?

The text has been modified and now reads Tekscan acquisition software allowed real-time visualisation of the COP trace and offline analysis.

Line 132. Does this mean that a minimum of 20 strides total were collected for each horse in walk but only 10 trot strides? Why?

Yes. For each horse we recorded two walk sequences and only one trot sequence; the single trot sequence gave us an adequate number of strides and at the same time meant that the lamer horses were not trotted excessively. 

Line 137. Here you say 501 points but under data analysis, it says 101 points. Am I missing something?

The Tekscan output for each stride contained a variable number of coordinates, depending on the stance time. To facilitate the calculation of the average COPp in Matlab, each COPp was first normalized to 501 points (this number was selected from the original number of points in each record to obtain a good fit) and then the average COPp was computed as the average of all strides point by point. In order to allow comparison with the Tekscan software output, the average COPp was finally interpolated to 51 points. 

For the Procrustes analysis, the COPp were interpolated to 101 points, to allow comparison with the study by Nauwelaerts et al. 

In summary normalisations were performed to allow visual comparison between Matlab calculated COPp and Tekscan software output, and between our data and previously published ones. This has now been better explained in the manuscript. Whether normalizing to 101 or to 501 points, no qualitative changes were noted.

Line 138. Why did you want to make this comparison? Also, the results for this comparison are not reported anywhere. 

We wanted to be sure to have comparable data as we had noticed that the average COPp calculated with Matlab did not fully match the Tekscan output. Being unaware of the averaging procedure used by the Tekscan software, we finally preferred to use our Matlab generated figures, as we knew exactly how they were calculated and allowed full reproducibility. 

Line 141. It’s not obvious why this normalization process was undertaken, given that normalization is inherent to the Procrustes algorithm. Was this for calculating the shape indices that you describe further below? Please specify for which analysis/analyses these normalized data were used.

The built-in Matlab function used to perform the Procrustes analysis requires as input two curves with an equal number of points (coordinates). For this reason, the different COPp had to be normalized a priori and the normalization process has been explained in the manuscript. For the Procrustes analysis a normalization to 101 points was done, based on the paper from Nauwelaerts et al. This normalization is described in the paragraph “Procrustes analysis”.

Similarly, for the determination of the average COPp, a constant number of points was required. For this reason a normalization to 501 points was performed, this is described in paragraph “Data processing”.

Page 14

Line 153. I interpret this that you are comparing strides that come from the same horse, hoof and gait. In that case, why was it relevant to use Procrustes for this? No scaling, translation or rotation should be necessary to compare the center of pressure paths in that case, unless the sensor position relative to the hoof was unstable. If that was not the case a simple root mean square difference should give you the same result. That is, you can apply the formula for calculating the D-value directly to the data without transforming it first. Please explain the rationale for using this analysis. Further, when comparing multiple measurements it is more appropriate to use generalized Procrustes analysis, which the Matlab function that you have used doesn’t do. Please explain why you chose this approach.

We understand your concern and try now to explain our rationale. Although it is true that no scaling, translation or rotation might be necessary to compare the COPp for the same horse, the Procrustes distances should converge to a simple root mean square difference in that case. In addition, as the Procrustes distances have been used as a mean of comparing not only different strides for the same horse/hoof/gait, but also in between different horses/hoof/gait, we believe that using the Procrustes distances among all comparisons is more consistent.

We agree that the generalized Procrustes analysis might be a better approach in some cases. However, for the comparison performed in this study, we do not expect any difference between the Procrustes analysis and the generalized Procrustes analysis, as all COPp are described using the same temporal sequence (i.e. starting from the footstrike and ending with the lift-off), and only 2 shapes at time were compared (I.e. stride 1 with stride 2, then stride 1 with stride 3, and so on)

Find 155. What did you input as target shape and comparison shape to the matlab procrustes function, respectively? Note that D-values returned from this function are normalised to the target shape. Please report this both for comparing strides from the same trial/hoof, and for comparing sound and lame limbs. Also, since it is known that chronic lameness can influence hoof shape over time, is it possible that this has confounded the D-values when comparing sound and lame limbs?

We always compared two COPp, for example stride 1 with stride 2, and from our results we did not observe any difference by comparing stride 1 (target shape) to stride 2 (comparison shape), or comparing stride 2 (target shape) to stride 1 (comparison shape). We did always both ways for all the strides and we always had the same output. For this reason, we do not see any need for explicitly specifying which shape has been considered as target shape and which one as comparison shape. This has now been specified in the manuscript. 

A modified hoof shape due to chronic pain could surely influence the similarity of COPp among strides, as an abnormal shape might allow only very limited footing patterns. This has been now added to the discussion as potential influencing factor.

Figure 1. The hoof print in this figure looks weirdly distorted, it seems that the width/height relationship is off. Also, please capitalize Cm in the same way you have done in table 2 and please indicate the unit of the X and Y axes.

Thanks for the comment we adapted the figure according to your comments. The main reason why it looked distorted is that the axis scale was slightly different (i.e. visually it appeared a bit squeezed in the horizontal direction). In the manuscript, the figure has now been reported using exactly the same scale for both axes. In addition, we specified that the units of the axes are centimetres. 

Line 163. I assume this should be table 2 rather than table 3?

Yes, thanks for noticing this error.

Page 15

Table 2. Isn’t Cm also illustrated in Figure 1?

Yes it is. Thank you for noticing. 

Line 172. Please specify how footstrike and lift-off events were defined and determined.

We based our definition of footstrike in the COPp image as the first appearing COP point, representing the hoof impact with the ground. Similarly, the lift off is the last COP point on the COPp, representing the last hoof contact with the ground. The Tekscan software provides animations in which hoof-prints and COP are shown in motion, thus the temporal sequence is visually easily recognizable. The numerical software outputs are the coordinates of the COP in temporal sequence during stance, thus first and last COP points are easily determined. This has now been specified in the text.

Page 16

Line 189. “were initially analysed for all 17 horses together and subsequently as independent variables” I don’t understand this part of the sentence. From reading the next sentences I assume this means that you made a group-level analysis ignoring that each horse has two limbs, and then a pairwise analysis taking this fact into account. I don’t quite understand why the first type of analysis makes sense in this context though, see my following comment. 

All comments regarding the statistics are answered together after the comment to Line 191. 

Line 190. The Mann Whitney U doesn’t take repeated measures into account. This means that this analysis can’t be used for this dataset since each horse contributes one value for the left and one value for the right limb (minimum, more if you don’t use a summary measure per limb). I suggest that you use a repeated measures ANOVA or a mixed model, that includes horse and limb nested in horse as random variables (assuming stride-by-stride data) and transform the outcome variable as needed to achieve normal distribution of the residuals.

All comments regarding the statistics are answered together after the comment to Line 191. 

Line 191. “To characterise the behaviour of the single-strides COPp and the resulting D-values in individual horses, a subject-based analysis was performed using the Wilcoxon Signed Rank test to compare limbs for each gait.” I’m not entirely sure what you have done here. From my perspective it makes the most sense to compare sound and lame limbs within each horse and across all horses, using a summary measure, e.g. the median D-value, for each horse and hoof. From the legend for figure 4 it seems that you have indeed done this. However, you have failed to describe what summary measure you used. From the results text, it seems that you have also applied this analysis on an individual level. That’s not exactly great since the sample size will depend on the number of strides recorded, and since consecutive strides hardly can be considered independent measures, which is an underlying assumption of the statistic.

Thanks for your understandable criticism and for the constructive suggestions. We have now completely modified this statistic section according to your suggestions including only a mixed linear model for comparing sound and lame, we agree that this is the best way to take into consideration all the factors at the same time. This surely strongly improves the quality of the manuscript and made it easier to understand for the reader. 

Figure 4 legend has been re-written according to your suggestion. 

Line 195. You might want to be consistent with how the signed rank test is capitalized or not. Compare this and the previous sentence.

Thank you for noticing. Corrected accordingly.

Page 18

Tables 3 and 4. I don’t understand how this is two tables. It would make more sense to make them table 3 A and B if you want to have a separate annotation for each panel.

Table 3 and 4 have now been removed, as we don’t present the individual horse analysis.

Line 224 cont. This statistical evaluation is not described in the statistics section. Please describe how this test was applied and what statistic that was used.

A signed rank test was carried out to compare the average D-values. The text has been adapted following your suggestion.

Page 19

Table 5. The table contains unexplained abbreviations. You may want to refer to table 3 for this. “From left to right: shape indices for lame and sound limbs deriving from the first walk trial (WALK1), from the second walk trial (WALK2) and from the trot trial (TROT).” Is there a specific reason why the two walk trials are reported separately in this table but not otherwise?

Thank you for the comments, we added explanations for the abbreviations. Regarding walk trials, we have combined the two sequences for calculating the D-values, but not for the figures, as here we represent the average COPp of a real recorded sequence, not a summary of 2 separated sequences. 

Supporting information. The Excel document contains one sheet with D-values, however, it isn’t specified to which horse each value belongs. Further, since there are a lot of missing values in the end of some columns but never further up, it seems that all values on the same row do not belong to the same horse. This means that these data can’t be used to reproduce your statistics. Please update this sheet to include horse information.

Done according to your suggestions

Reference list. A few of the references are in correctly formatted or like information, e.g. name of the journal, e.g. nr 3 and 28.

References have now been corrected in the definitive manuscript version.

Answers to Reviewer 2

Dear reviewer, many thanks for your thorough evaluation of our work and the resulting constructive comments. They are all valuable, we have tried to implement as many as possible and this will certainly improve our publication.

Reviewer #2: This paper seeks to look at COP in horses with chronic OA.

Title: if definitive localization of the lameness via nerve or joint blocks was not performed the title has to change to just indicate lameness and not lameness due to chronic oa.

We would like to keep “chronic osteoarthritis” in the title. The need for diagnostic analgesia prior to inclusion in the study was deemed redundant when we looked at the sum of the information available to us to confirm the cause of lameness as osteoarthritis. All the horses underwent diagnostic analgesia (repeatedly) as part of their (often prolonged) history by their private veterinarians. We realized that this was not clearly stated in our previous manuscript version. Indeed, the diagnosis of OA was “the criterium” for inclusion (we looked for horses with a diagnosis of OA). Our clinical evaluation, lameness evaluation and radiographs all supported and confirmed the previous diagnosis. We acknowledge that it is a limitation not to have repeated the diagnostic analgesia but hope that the evidence we present is strong enough to be able to keep the title as is. 

General Abstract Comments:

ideally indicate the number each of fore and hind limb lamenesses and their sites.

Are these bilateral lamenesses (as chronic oa tends to be)? Aka the “sound” side may still be affected. How were they diagnosed as chronic OA?

I am a little concerned about lumping different lameness and sites and front and hind limbs together.

Table 1 shows the distribution of fore and hind limb lamenesses. It also shows the source of the lameness. Although all horses included in the study showed a unilateral lameness, a number of horses (9) had findings that meant a bilateral lameness could not be excluded. All horses had a history of lameness of at least 3 months duration (chronic OA). A statement to this effect is now included in the text. We agree with the reviewers concerns with regard to lumping lamenesses together, but we were pleased to be able to recruit this many osteoarthritic cases in a relatively short space of time. Being more specific with inclusion criteria would have made the study very difficult to carry out. 

Line 42- is the lateral footstrike on all limbs or only the lame ones?

All measured limbs showed this pattern of landing. We have modified the sentence to clarify this point. 

40-41- p values? Quantification of the difference? Statistical tests used?

We have now included the statistical tests and P- Values in the text.

44 by typical do you mean unique to each horse? Typical makes it sounds as if all horses have the same response, which the second part of that sentence doesn’t support. Highly repetitive within a horse correct?

Yes by typical we mean unique to each horse and highly repetitive. We have now used the word characteristic rather than typical since it seems more appropriate.

INTRODUCTION

60. could you clarify this phrase? Is consistency the best word? What is this in relation to? Objective lameness evaluation? Only COP measurement? Do you have concerns about achieving this in lame horses? It probably needs a little better elaboration, I will rewrite it to make it clearer.

This sentence has now been altered to read; ‘The prerequisite is, however, stride-to-stride repeatability to ensure reliable data’. This should clarify this sentence. Of course performing initial evaluations to recognize helpful patterns is not ideal in lame horses. A study looking at COPp in sound horses would definitely provide more robust data as a starting point in pattern recognition. We did not have that here and it is a limitation. This is included in the limitations section. 

66 would single stride data be more variable and not as representative as averaging several strides? How do you “pick” the stride? Potential bias from that process- please justify further

By solely assessing an average value, the stride-to-stride variability within a trial is unknown. We wanted to compare single stride data in order to recognize how repeatable strides were within a trial (rather than just looking at the average). 

It was not our intention to suggest evaluating single stride in the future but just to asses that single- stride variability in between a trial could provide variable additional information to the average analysis that is commonly performed. We adapted the manuscript accordingly. 

74- please add into the intro the literature from which you based your hypothesis that the COPp would be more consistent in lame horses.

This hypothesis is based on the results of several publications in human medicine These studies reported that orthopaedic pathology has an influence on COP by reducing its variability. The publications are listed below and have now been included in the text. 

• Onuma R, Masuda T, Hoshi F, Matsuda T, Sakai T, Okawa A, Jinno T. Measurements of the centre of pressure of individual legs reveal new characteristics of reduced anticipatory postural adjustments during gait initiation in patients with post-stroke hemiplegia. J Rehabil Med. 2021 Jul 2;53(7):jrm00211. doi: 10.2340/16501977-2856 . PMID: 34159392 ; PMCID: PMC8669160.

• Wong AM, Pei YC, Hong WH, Chung CY, Lau YC, Chen CP. Foot contact pattern analysis in hemiplegic stroke patients: an implication for neurologic status determination. Arch Phys Med Rehabil. 2004 Oct;85(10):1625-30. doi: 10.1016/j.apmr.2003.11.039 . PMID: 15468022 .

• Donker SF, Roerdink M, Greven AJ, Beek PJ. Regularity of center-of-pressure trajectories depends on the amount of attention invested in postural control. Exp Brain Res. 2007 Jul;181(1):1-11. doi: 10.1007/s00221-007-0905-4 . Epub 2007 Mar 31. PMID: 17401553 ; PMCID: PMC1914290.

77 difference between front and hind limbs? Sound and lame limbs? Limbs on different horses?

This has now been clarified to read; average COPps would be consistent within a limb for subsequent trials but different when compared to opposite limbs 

79 do you mean objective lameness severity? Positively or negatively correlate? Please clarify

This now reads; COPp shape indices would differ between sound and lame limbs, and the difference would become increasingly evident with increasing lameness severity (as recorded both subjectively and objectively). This hypothesis was based on a publication by Lopez et al.. This study recorded COP in dogs and found a more cranial start to the COP in the lame limb. 

METHODS

91- radiographic presence of OA does not necessarily correlate to clinical lameness. Were these horses blocked (nerve or IA) to prove that the OA was the source of the lameness vs. soft tissue origin? How was the site of lameness localized? Without this information, you can state you studied COP in lame legs, but I don’t think you can positively state that it was chronic OA related. This may require a title change subsequently.

All of the horses were already diagnosed with chronic OA by other vets prior to inclusion in the study and were put forward for that reason. The combination of a reliable history, our clinical evaluation, subjective and objective lameness evaluation and radiographic evidence confirmed OA as the cause of the lameness. As mentioned before, repeating diagnostic analgesia was not in the interest of the horses included in the study and was deemed not necessary. This has now be added in the limitations section. 

95 many of these horses are retired, how can you then use inclusion criteria of decreased performance or intolerance to being ridden?

This was one of many criteria used to assess the clinical state of the study horses. Others included: persistent lameness, abnormal gait, difficulty lying down or standing up. We have modified the manuscript to avoid confusion. 

Table 1- subjective lameness grade should be referenced and explained. Was there a inclusion criteria for subjective lameness?

The lameness score is the 0-10 scale recommended for use in the UK. It is explained in detail and referenced in the chapter "Lameness scoring and gait symmetry analysis". We have included the grading scale in the table. No inclusion criteria for subjective lameness was explicitly stated.

Table 1- I appreciate your indicating bilateral or not for affected limbs, how did you assess the issue with the other limb?

In order to be able to incorporate horses with a history suggestive of bilateral lameness in the study we designated these horses as having a ‘most lame limb’. These horses had a leg that was both subjectively and objectively assessed as being worse than the opposite. While not perfect, this enabled inclusion of these horses for assessment of their COPp including comparison between the ‘good’ (sound) and the ‘bad’(lame) leg. Our approach is described in the text. Just how much a bilateral lameness influenced the horse at the outset of the study and its influence on COP would be worthy of closer attention but was not considered here. Nevertheless, potential presence of bilateral lameness was added as a covariate in the newly performed linear mixed effects model.

 Did you compare COP’s to the other potentially affected limb in these instance or? Aka what did you use for “sound” comparisons? 

Yes, as stated above the less lame limb was taken as a reference for comparison with the “most lame limb”. Potentially because the difference in lameness between opposite limbs was substantial, patterns obtained in those horses suggestive of having bilateral lameness did not differ from those obtained from truly unilaterally lame horses. 

How did you decide you could pool together front and hind limbs? GRF in front vs hind limbs are different and COPs likely will reflect this too. Is there a refernce which has pooled front and hind together before validating this? 

GRF is an absolute value. We were hoping to recognize patterns of COP and how they change within a trial, or between trials, and their comparison between limbs and with pathology. Mixing front and hind limbs is not ideal and subtle differences may go unnoticed. However since the majority of the comparisons were within a hoof over time, or between left and right this does not seem relevant to our study. There are only very few publications looking at COP and COPp in horses so a reference which pools front and hind limbs together is lacking. 

Who assessed the subjective lameness? One person? Many? Intraclass correlations for that if many? Can you describe how they were assessed? Footing? Circles? Etc? at the same farm or ? What are the qualifications of the vet/s who diagnosed them?

All subjective lameness examination was done by one veterinarian with several years of experience in the field of equine medicine (Dr. med. vet Sabrina Andreis (FVH Horses, swiss specialization Program for equine medicine), co-author in this paper). The horses were assessed at a walk and trot (solid surface) on a straight line as well as on a circle at walk. These important details have now been added.

97- corticosteroids can have anti-inflammatory effects that last longer than 4 weeks- if pain returns, it is suggestive of a soft tissue component to the issue or a misdiagnosis of the joint that is painful, please discuss/explain

We are in agreement with the statement about the duration of effect of corticosteroids. The 4 week time-frame was set as minimum for the purposes of the primary study, as we could not pretend horses with OA to be untreated for too long time. In humans, it is known that OA pain can in large part derive from periarticular structures, including soft tissues. As at the time of inclusion in the study horses showed clear lameness, we knew that the effect of possible previous treatments was at least partially gone, and this was considered enough for the purpose of the present study. 

99 was blood used to look for other medical issues? Clarify please if this related to part of your exclusion of other medical issues indicated in the first part of the sentence. As it is written now it sounds as if it is something else and not for that reason.

Blood was taken to assess the health status of the horses for the main study. It is not relevant to this study so that we have removed this sentence. 

100 17 horses out of how many? How were they located? Previous clients? Elicited via an email for the research?

72 horse owners responded to a posting in the social media. 20 horses were considered eligible and were invited for a complete clinical examination. Finally 17 met inclusion criteria. We have added this information in the text. 

106- qualifications? One vet only?

Dr. med. Vet. Sabrina Andreis (FVH Horses, swiss specialization Program for Equine medicine), co-author of this paper and a veterinarian with many years of experience in equine medicine, examined all the horses. This has now been included in the text.

107 foreseen is not the correct word here, maybe just delete? 

We have changed the sentence, thank you. 

109 may be an aberrant | after lameness? I’d like to see this paragraph above the table as this explains the subjective lameness grades. I think there should be an exclusion on the high end of the lameness scale for this study, no? 

Thank you for this comment, indeed we did not establish any explicit exclusion criteria for lameness grading. In Switzerland, it would be highly unlikely to see chronically affected horses with a lameness score ≥8, as this would be considered unethical. The maximal lameness score that we recorded was 7 (Table 1). 

119 Do you have references that you can cite that indicate that if bilateral lameness is present that the Xsens will always pick up on it in horses that aren’t blocked? According to Keegan et al. AJVR 2012 “By contrast, the inertial sensor system measures the asymmetry of head movement between the right and left portions of a stride. Horses with bilateral forelimb lameness, in which the severity of lameness in opposite limbs is equivalent in every stride, would be judged as symmetric if inertial sensors were used…but the inertial sensor system cannot measure the absolute severity of lameness in 1 limb in any 1 stride.” So that the inertial sensors can pick out the lamer of paired legs only. This needs to be further detailed/justified/supported if you are claiming that the opposite leg serves as a control based on xsens inertial sensor data.

Also how did, or did you address/recognize compensatory lameness?

Our horses were clearly asymmetric in the subjective as well as in the Xsens evaluation. Our goal was not to quantify the lameness with the Xsens but just to confirm the “most lame limb” for which horse was diagnosed already based on the reporting from the private vets and the subjective lameness evaluation. We added in the table of the demographic data also the values from the Xsens.

119- I don’t know if this journal allows you to refer to items by brand names (equigait)

Thank you for noticing this, we have removed the brand name. 

126 bilateral lameness, especially when there is evidence for it in the history, ideally would have had see the horse’s lamest leg blocked to show that the opposite leg didn’t become sore.

The issue of potential bilateral lameness has been addressed in a previous answer. For the present study no diagnostic analgesia was repeated and what we define the lame limb is indeed simply the “lamest” limb.

131 what did you do to target a constant speed? Did you have markers set up to measure acceleration or deceleration?

Speed of walking or trotting was not standardized, we did try to obtain a similar gait speed/impulsion as required for a “subjective” lameness examination but nothing more. To obtain a constant speed several attempts and recordings are generally required, and we preferred to avoid such a strain for the horses and the people involved (most of the time, it was the owner him/herself to lead the horse during the examination).

191-193 what did you do to correct for multiple comparisons?

According to your and the other reveiwer’s criticism, we have completely redone the statistical analysis for this part of the data. In the new submission we present now a mixed model analysis, that take into account the individual horse as random effect and limb (front versus hind) and suspicion of bilateral pathology as covariates. 

204 have you looked at front alone and hind alone? I am leary of pooling front and hind together or would like to see the numbers and stats that you have to demonstrate they are ok to be pooled.

Front versus hind lameness have now been considered as covariates in the mixed model. The results of this analysis indicate that this factor does not influence the outcome, but surely a larger sample would be necessary to confirm these preliminary results. 

214 please clarify significant and trend? Trend normally indicates a non-significant finding so these appear contradictory.

Thank you for your remark, we simply intended to highlight the fact that horses having significant lower D values for the lame limb at walk, showed the same at trot. We now removed all the sentences about the individual analysis, which was highly criticized.

Table 3 and 4, are these corrected p values in any way for the number of comparisons indicated? Also this is confusing as the p value looks to be just looking at within a horse between the lame and sound leg? But I think you compared just overall lame to sound? Are some of these p values repeated as WIlcoxin should take into account all lame vs all sound. Some of these p values may reflect type 1 errors and become non significant if adjusted for the number of tests.

Our previous answer regarding the statistical approach should now have clarified the issue.

Table 6 define Q, clarify division by left right, include N=17 describe how this related to lateral sidedness.

Thank you for your comment. Q stays for Quadrant (this is now in the Table legend) and we adapted the table to make it clearer. 

230 p values ofter just go to one or 2 digits beyone the . sometimes you are 2 and other times 3, please see journal conventions

Thanks for noticing this incongruence, now all P values are reported with 3 digits. 

DISCUSSION

246-250 this is too intro like, please address whether you accept, partially accept or reject your specific hypotheses briefly and concisely, then go into details comparing to lit in later paragraphs.

Paragraph 255 too intro and method like, need to compare your results to the literature as the focus.

Thank you for your suggestion. Nevertheless, we try now to explain why we would like to keep our original structure, reminding the reader the aims first and then addressing each hypothesis (and whether it should be accepted/rejected) in following separate discussion sections. We believe that this should help to remind the different goals and to better understand the discussion of findings. We had 3 main hypotheses, providing a list of accepted/rejected hypotheses in the beginning would result in a rather boring start of discussion and we would be obliged to repeat each topic anyway later in the text. For each section/issue, a comparison of our results with the available literature is presented, this might seems poor as in several cases not much evidence has been published so far for the topic. 

269 decreases in variability… is higher…. Is unclear. Please clarify

Here it is meant that the decrease in variability was stronger and more important at trot for the pool data analysis. We now present only the mixed model results, for both walk and trot similar P values have been found. This sentence has been deleted.

277 phenotyped how? I’m still uncomfortable claiming the pain is definitively OA pain, need to mention different joints can express things differently in the limitations if not here more explicitly

This part of the text has now been modified and hopefully made clearer.

278 at the individual what? I would like to see statistics behind this claim that it wouldn’t affect things. I particularly need to see data of front vs hind, to understand if they can legitimately be pooled.

This factor is now considered as a covariate in the mixed model analysis.

280- lack of diagnostic analgesia to localize the lameness should be explicit earlier and in the limitations. Given this information, I think you have to rephrase the paper in light of lameness and not chronic oa. You can include in the chart areas of previous concern from oa, and should include historical data of the length of lameness in the leg (over the short and long term).

We hope that the provided explanations help now to reassure the reviewer about this issue. The text has been modified accordingly whenever possible (inclusion criteria, study limitations).

305-306 too method like, delete

Thank you for the comment, we removed the sentence. 

308 this is somehow not surprising, is not clear please rephrase

Rephrased

315 I dislike tendency, if this was NOT statistically significant it should not be discussed as if it is, please remove and just say this wasn’t significantly clear in this study but previous studies found….

Thank you for the comment, now only the results of the mixed model are discussed

328 Nauwelarts didn’t observe this study please rephrase, just say Similar to the findings in X, the present study confirmed…

Thanks a lot for the comment, the text has been modified. 

330 confirm if this is lame and sound legs or just one of these 

We actually observed for both (lame and sound limbs) that the COPp is highly repetitive. The text should now be clearer. 

331-332 remove this sentences as this wasn’t significant There seems to be a tendency for even

331 lower COPp variability in the presence of a painful limb pathology, possibly indicating a preferential 332 way of hoof loading to decrease discomfort

See comments above

337 limitations- does this journal request this to be last? Normally this is before conclusions.

Limitations are now reported before conclusions. Thanks for noticing this. 

341 lameness and pain free is awkward, please rephrase

Done

346 -347 what references are these? Please cite them here and above

This comment is based on following paper: Pfau T, Boultbee H, Davis H, Walker A, Rhodin M. Agreement between two inertial sensor gait analysis systems for lameness examinations in horses. Equine Vet Educ 2016;28(4):203–8

It is already cited in the method section already, we will add the citation here. 

350 please indicate who was shod or not in the table

Shoeing is reported in Table 1, for some reasons some columns went lost while uploading the file. Now the table should be complete.

351 references for farriery? This seems to be appropriate to work into other aspects of the discussion

Farriery issues are now mentioned in the final part of the discussion

356-358 this sentence is unclear, please simplify

The idea behind this is that if both hooves are equally shod or not shod, we have modified the sentence to make it clearer. 

References

Journal names seem differently referenced, AJVR and others are totally spelled out while Equine Vet J is partially shortened as are others, please fix per journals standard

References have been checked and adapted to the journal standards, the layout was provided by the reference manager software.

---

## [Decision Letter · Decision Letter 1]

21 Aug 2023

PONE-D-23-12780R1Evaluation of the hoof centre-of-pressure path in horses affected by chronic osteoarthritic painPLOS ONE

Dear Dr. Buser,

Thank you for submitting your manuscript to PLOS ONE. After careful consideration, we feel that it has merit but does not fully meet PLOS ONE’s publication criteria as it currently stands. Therefore, we invite you to submit a revised version of the manuscript that addresses the points raised during the review process.

In particular, those related to calibration and potential effects on data and data interpretation raised and clarified by the reviewer below. 

We look forward to receiving your revised manuscript.

Kind regards,

Aliah Faisal Shaheen

Academic Editor

PLOS ONE

Reviewers' comments:

Reviewer's Responses to Questions

**Comments to the Author**

1. If the authors have adequately addressed your comments raised in a previous round of review and you feel that this manuscript is now acceptable for publication, you may indicate that here to bypass the “Comments to the Author” section, enter your conflict of interest statement in the “Confidential to Editor” section, and submit your "Accept" recommendation.

Reviewer #1: (No Response)

2. Is the manuscript technically sound, and do the data support the conclusions?

Reviewer #1: Yes

3. Has the statistical analysis been performed appropriately and rigorously? 

Reviewer #1: Yes

4. Have the authors made all data underlying the findings in their manuscript fully available?

Reviewer #1: Yes

5. Is the manuscript presented in an intelligible fashion and written in standard English?

Reviewer #1: Yes

6. Review Comments to the Author

Reviewer #1: Thank you for attending to my comments on the previous version of the manuscript. The revised version is substantially improved. However, I have a few additional comments, see below.

Regarding calibration of the measurement equipment, my concern for lack of validation was referring to the pressure measurement equipment. The inertial measurement unit system used for assessment of movement asymmetries I don’t have any issues with. Sorry for being unclear.

Regarding calibration of the pressure measurement equipment, this is actually relevant even if you are only interested in the pressure distribution. The reason for this is that the output from the sensor isn’t linear with respect to the applied force, which means that the relative pressure distribution will be more accurate if calculated from calibrated values rather than raw voltages. This conference paper shows a graph nicely illustrating the nonlinearity of Tekscan sensors:

https://isbweb.org/images/conf/2005/abstracts/0263.pdf

Apart from calibration, Tekscan manual also recommends that you perform equilibration of the sensor. This is because the voltage output from each sensing element isn’t perfectly the same. In my experience, this is not a huge issue with a completely new sensor. However, with repeated use sensing elements that are subjected to high pressures become less sensitive to pressure over time. But I assume that the sensors you used were single use only?

Thank you for adding the movement asymmetry values to table 1. This clarifies my question of how the subjective and objective data were combined as both lameness degree and objective data are now available in this table.

The description of the first Procrustes analysis in the methods section is still unclear to me. When you compare the shape of the single-stride COPp, I assume this is comparing strides for the same limb within the same measurement sequence? Please add this in the text to avoid any confusion. Further, you describe that you compare stride 1 to stride 2 and state that “to avoid measurement errors, the Procrustes analysis was always performed between COPp recorded during 2 strides”. If you want to study within-measurement variation, then you would want to compare all pairs of strides, basically the Cartesian product. Please add a motivation why you were only interested in differences between consecutive strides (stride 1 vs 2, 2 vs 3 and so on). Maybe you’re trying to say in the sentence cited above that you anticipated that measurement errors would be more influential if comparing strides further apart (due to for example small shifts of the sensor position relative to the hoof during the trial)?

The statistical analysis and the description in the statistics section are much improved from the previous version. Thank you for this. Please add whether repetition was included as a linear or as a categorical effect as both are possible/reasonable choices (depending on the data).

In the results section, when reporting the results from the linear mixed model please add the estimated marginal effects in addition to the P values.

7. PLOS authors have the option to publish the peer review history of their article (what does this mean?). If published, this will include your full peer review and any attached files.

Reviewer #1: No

---

## [Author Response · Author response to Decision Letter 1]

28 Aug 2023

Reviewer #1: Thank you for attending to my comments on the previous version of the manuscript. The revised version is substantially improved. However, I have a few additional comments, see below.

Dear reviewer, thank you very much for considering our manuscript again and giving us further suggestions for improvement, we are very grateful for this. We will try to remove the last ambiguities in order to improve the manuscript even further. 

Regarding calibration of the measurement equipment, my concern for lack of validation was referring to the pressure measurement equipment. The inertial measurement unit system used for assessment of movement asymmetries I don’t have any issues with. Sorry for being unclear.

Regarding calibration of the pressure measurement equipment, this is actually relevant even if you are only interested in the pressure distribution. The reason for this is that the output from the sensor isn’t linear with respect to the applied force, which means that the relative pressure distribution will be more accurate if calculated from calibrated values rather than raw voltages. This conference paper shows a graph nicely illustrating the nonlinearity of Tekscan sensors:

https://isbweb.org/images/conf/2005/abstracts/0263.pdf

Apart from calibration, Tekscan manual also recommends that you perform equilibration of the sensor. This is because the voltage output from each sensing element isn’t perfectly the same. In my experience, this is not a huge issue with a completely new sensor. However, with repeated use sensing elements that are subjected to high pressures become less sensitive to pressure over time. But I assume that the sensors you used were single use only?

Thank you for addressing this important issue. Indeed, right from the beginning, it induced a lot of thoughts from our side even if in the manuscript we did not present and discuss it properly. We correct the method section, add a sentence in the limitations and will explain here below our original thoughts and considerations.

When we started using the Tekscan in field situations, we got the suggestion to use the walk calibration method from researchers who previously used the sensors in horses. In their published papers, the calibration method was never really described in detail, it was just mentioned that the calibration was performed according to manufacturer recommendations. We soon realised that there is missing knowledge around this issue, that there are no generally accepted methods for field use and that the walk calibration is not adequate for accurate pressure measurements in case of gait asymmetries. While indeed we conditioned the sensors as suggested and calibrated them before data recordings using walk calibration on each individual horse, we then went back to uncalibrated force output to describe pressure distribution, as you know from our current manuscript. In the course of our literature search, we came across the Brimacombe paper (the full publication of the abstract you also mentioned), and once more, this convinced us that the calibration methods have a large role in the variability and errors of the pressure displayed by the Tekscan when compared to loads created from a materials testing machine. The authors suggest recording the force data as raw (uncalibrated) values and running a user-defined calibration algorithm offline, adequate for the range of pressures and experimental protocol used.

To make clear to the readers that we were not interested in absolute pressure data (but only in relative data, as provided by the Raw Sum display of the Tekscan), we stated that “we used an uncalibrated pressure measurement system”, which indeed is not correct, we merely used uncalibrated data for the purpose of the study. This sentence is now changed and we add the conditioning and walk calibration procedure in the description of the methods.

As you say, what we should have done to render the raw cells output perfectly homogeneous and to account for the small variation possibly existing among sensing elements would have been the equilibration procedure. For this purpose, a special device able to apply a very uniform pressure on the whole sensing surface is needed. We did not have it available at the time of data recording and we are convinced that the potential effect of this slight cells output variability on the COPp behaviour can be considered negligible. Indeed, COP derives from the sum of all forces, small variations among cells would be thus easily “compensated” and not further amplified.

Literature: Effect of Calibration Method on Tekscan Sensor Accuracy, Brimacombe et al 2009. DOI: 10.1115/1.3005165

Thank you for adding the movement asymmetry values to table 1. This clarifies my question of how the subjective and objective data were combined as both lameness degree and objective data are now available in this table. 

Thank you for bringing this to our attention.

The description of the first Procrustes analysis in the methods section is still unclear to me. When you compare the shape of the single-stride COPp, I assume this is comparing strides for the same limb within the same measurement sequence? Please add this in the text to avoid any confusion. Further, you describe that you compare stride 1 to stride 2 and state that “to avoid measurement errors, the Procrustes analysis was always performed between COPp recorded during 2 strides”. If you want to study within-measurement variation, then you would want to compare all pairs of strides, basically the Cartesian product. Please add a motivation why you were only interested in differences between consecutive strides (stride 1 vs 2, 2 vs 3 and so on). Maybe you’re trying to say in the sentence cited above that you anticipated that measurement errors would be more influential if comparing strides further apart (due to for example small shifts of the sensor position relative to the hoof during the trial)?

Thank you for your comment. Regarding the Procrustes analysis of the individual strides, we will change it in the text. Regarding the comparison between strides, thank you for pointing out that it is not clear. We have indeed compared all the strides with each other and not just the consecutive ones. We realised that this was not written clearly enough, and we will change it accordingly in the manuscript.

The statistical analysis and the description in the statistics section are much improved from the previous version. Thank you for this. Please add whether repetition was included as a linear or as a categorical effect as both are possible/reasonable choices (depending on the data).

Repetition was included as a categorical effect because there were no different time periods to consider. 

In the results section, when reporting the results from the linear mixed model please add the estimated marginal effects in addition to the P values.

We added this in the manuscript.

---

## [Decision Letter · Decision Letter 2]

4 Sep 2023

Evaluation of the hoof centre-of-pressure path in horses affected by chronic osteoarthritic pain

PONE-D-23-12780R2

Dear Dr. Buser,

We’re pleased to inform you that your manuscript has been judged scientifically suitable for publication and will be formally accepted for publication once it meets all outstanding technical requirements.

Kind regards,

Aliah Faisal Shaheen

Academic Editor

PLOS ONE

Additional Editor Comments (optional):

Reviewers' comments:

Reviewer's Responses to Questions

**Comments to the Author**

1. If the authors have adequately addressed your comments raised in a previous round of review and you feel that this manuscript is now acceptable for publication, you may indicate that here to bypass the “Comments to the Author” section, enter your conflict of interest statement in the “Confidential to Editor” section, and submit your "Accept" recommendation.

Reviewer #1: All comments have been addressed

2. Is the manuscript technically sound, and do the data support the conclusions?

Reviewer #1: (No Response)

3. Has the statistical analysis been performed appropriately and rigorously? 

Reviewer #1: (No Response)

4. Have the authors made all data underlying the findings in their manuscript fully available?

Reviewer #1: (No Response)

5. Is the manuscript presented in an intelligible fashion and written in standard English?

Reviewer #1: (No Response)

6. Review Comments to the Author

Reviewer #1: (No Response)

7. PLOS authors have the option to publish the peer review history of their article (what does this mean?). If published, this will include your full peer review and any attached files.

Reviewer #1: No

---

## [Editor Report · Acceptance letter]

7 Sep 2023

PONE-D-23-12780R2 

Evaluation of the hoof centre-of-pressure path in horses affected by chronic osteoarthritic pain 

Dear Dr. Buser:

I'm pleased to inform you that your manuscript has been deemed suitable for publication in PLOS ONE. Congratulations! Your manuscript is now with our production department. 

Kind regards, 

on behalf of

Dr. Aliah Faisal Shaheen 

Academic Editor

PLOS ONE